# Stochastically Capturing Partial relationship among Features for Multivariate Forecasting

## Abstract

When tackling forecasting problems that involve multiple time-series features, existing methods for capturing inter-feature information typically fall into three categories: complete-multivariate, partial-multivariate, and univariate. Complete-multivariate methods compute relationships among the entire set of features, whereas univariate cases ignore inter-feature information altogether. In contrast to these two, partial-multivariate methods group features into clusters and capture inter-feature relationships within each cluster. However, existing partial-multivariate methods deal only with specific cases where there is a single way of grouping so once the grouping way is selected, it remains unchanged. Therefore, we introduce a generalized version of partial-multivariate methods where grouping ways are sampled stochastically (called *stochastic partial-multivariate methods*), which can incorporate the deterministic cases using Dirac delta distributions. We propose SPMformer, a Transformer-based stochastic partial-multivariate model, with its training algorithm. We demonstrate that SPMformer outperforms various complete-multivariate, deterministic partial-multivariate, and univariate models in various forecasting tasks (long-term, short-term, and probabilistic forecasting), providing a theoretical rationale and empirical analysis for its superiority. Additionally, by proposing an inference method leveraging the inherent stochasticity in SPMformer, the forecasting accuracy is further enhanced. Finally, we highlight other advantages of SPMformer: efficiency and robustness under missing features.

## 1 Introduction

*Time-series forecasting* is a fundamental machine learning task that aims to predict future events based on past observations, requiring to capture temporal dynamics. A forecasting problem often includes interrelated multiple variables (*e.g.*, multiple market values in stock price forecasting). For decades, the forecasting task with multiple time-series features has been of great importance in various applications such as health care (Nguyen et al., 2021; Jones et al., 2009), meteorology (Sanhudo et al., 2021; Angryk et al., 2020), and finance (Qiu et al., 2020; Mehtab & Sen, 2021).

For this problem, there have been developed a number of methods, including linear models (Chen et al., 2023; Zeng et al., 2022), state-space models (Liang et al., 2024; Gu et al., 2022), recurrent neural networks (RNNs) (Lin et al., 2023; Du et al., 2021), convolution neural networks (CNNs) (Wang et al., 2023; Liu et al., 2022a), and Transformers (Zhou et al., 2021; Liu et al., 2022b). These methods are typically categorized based on how they capture inter-feature information, falling into three types: (i) univariate, (ii) partial-multivariate, and (iii) complete-multivariate methods. Univariate methods capture only temporal dependencies, while complete-multivariate methods incorporate additional modules to account for complete dependencies among all the given features. In contrast, partial-multivariate methods divide the feature set into multiple subsets and capture dependencies within each subset. The differences between the three methods are illustrated in Figure 1.

Partial-multivariate methods, which focus on relationships among mutually significant features, can enhance performance by excluding insignificant features that may act as noise. However, these methods often face limitations due to their deterministic approach to grouping (Aguiar et al., 2022; Pathak et al., 2021). Once optimal clusters are determined through some specific procedures, they remain fixed throughout training and inference. This rigidity fails to accommodate more complex scenarios where features may be grouped in various ways. For instance, stock prices might be

Figure 1: Visualization of three types of methods where $S$ is the size of each cluster (subset) in partial-multivariate methods. In partial-multivariate methods, traditional approaches adhere to a single grouping strategy throughout training and inference. In contrast, we propose to introduce variability by stochastically sampling from all possible grouping configurations.

categorized by different criteria such as market capitalization (e.g., inclusion in the S&P 500), industry sectors (like financials, healthcare, or energy), or geographic regions.

To address this limitation, we introduce a generalized version of partial-multivariate methods called *Stochastic Partial-Multivariate* methods. In this approach, a grouping way is not fixed but sampled stochastically—note that deterministic methods can be included by setting the sampling distributions of the grouping strategies to distributions such as Dirac delta. To implement this concept, we propose Stochastic Partial-Multivariate Transformer, SPMformer. Inspired by Nie et al. (2023), SPMformer is capable of capturing any partial relationship through a shared Transformer by individually tokenizing features and computing attention maps for selected features. Additionally, we introduce a basic form of training algorithms for SPMformer that are based on random sampling, under a usual assumption that the prior knowledge on how to group features into subsets is unavailable.

In experiments, we demonstrate that SPMformer outperforms existing complete-multivariate, (deterministic) partial-multivariate, or univariate models in various forecasting tasks including long-term, short-term, and probabilistic forecasting. To explain the superiority of our stochastic partial-multivariate method against the other methods, we provide a theoretical analysis based on McAllester's bound on generalization errors (McAllester, 1999) with supporting empirical analyses. To further enhance forecasting performance, we introduce a simple inference technique that leverages the inherent stochasticity of stochastic partial-multivariate methods. Finally, we show other useful properties of SPMformer: efficient inter-feature attention costs against other Transformers including inter-feature attention modules, and robustness under missing features compared to complete-multivariate models. To sum up, our contributions are summarized as follows:

- We introduce the novel concept of *Stochastic Partial-Multivariate* methods in the realm of time-series forecasting, generalizing existing forecasting models. To realize this concept, we develop the Transformer-based SPMformer along with its training algorithm.

- Our extensive experimental results demonstrate that SPMformer outperforms recent baselines across various forecasting tasks, including long-term, short-term, and probabilistic forecasting. We also provide a theoretical analysis to substantiate the superiority of our model, supported by empirical evidence.

- We propose an inference technique for SPMformer that further enhances forecasting accuracy by leveraging the inherent stochasticity of stochastic partial-multivariate methods. Additionally, we identify several advantageous properties of SPMformer compared to complete-multivariate models, including efficient inter-feature attention costs and robustness in scenarios with missing features.

## 2 RELATED WORKS

**Complete-multivariate and univariate methods.** To solve the forecasting problem with multiple features, it is important to discover not only temporal but also inter-feature relationships. As for inter-feature relationships, some existing studies often aim to capture full dependencies among a complete set of features, which we call complete-multivariate methods. For example, some approaches encode all features into a single hidden vector, which is then decoded back into feature spaces after

Figure 2: Architecture of Stochastic Partial-Multivariate Transformer (SPMformer). To emphasize row-wise attention operations, we enclose each row within bold frames before feeding them into the attention modules. In this figure, the subset size $S$ is 3.

some processes. This technique has been applied to various architectures, including RNNs (Che et al., 2016), CNNs (Bai et al., 2018), state-space models (Gu et al., 2022), and Transformers (Wu et al., 2022). Conversely, other complete-multivariate studies have developed modules to explicitly capture these relationships. For instance, Zhang & Yan (2023) computes $D \times D$ attention matrices among $D$ features by encoding each feature into a separate token, while Wu et al. (2020) utilizes graph neural networks with graphs of inter-feature relationships. Additionally, Chen et al. (2023) parameterizes a weight matrix $W \in \mathbb{R}^{D \times D}$, where each element in the $i$-th row and $j$-th column represents the relationship between the $i$-th and $j$-th features. Unlike complete-multivariate methods which fully employ inter-feature information, new methods have recently been developed: univariate methods. (Zeng et al., 2022; Xu et al., 2024; Nie et al., 2023; Wang et al., 2024; Lee et al., 2024) These methods capture temporal dynamics but ignore the inter-feature information by processing each of $D$ features separately as independent inputs.

**Deterministic partial-multivariate methods.** Unlike complete-multivariate and univariate methods, partial-multivariate methods aim to capture partial relationships among features by grouping them into various subgroups and computing relationships within these subgroups. Traditional partial-multivariate methods typically search for a single optimal grouping method, which is then applied consistently throughout all training or inference stages; these are referred to as deterministic methods. For instance, in Pathak et al. (2021), the optimal grouping is determined using truncated SVD and K-means clustering, with different autoregressive models assigned to each cluster. Conversely, Aguiar et al. (2022) introduced training-based methods that simultaneously tackle prediction and clustering tasks to identify an optimal grouping. In contrast to these approaches, we propose a fundamentally different method for capturing partial relationships among features: we maintain the grouping method as stochastic rather than fixed, offering a more flexible modeling strategy.

## 3 METHOD

### 3.1 STOCHASTIC PARTIAL-MULTIVARIATE FORECASTING MODEL

In this section, we provide the formulation of the stochastic partial-multivariate forecasting model. To simplify a notation, we denote the set of integers from $N$ to $M$ (inclusive of $N$ and exclusive of $M$) as $[N : M]$ (i.e., $[N : M] \coloneqq \{N, N+1, \ldots, M-1\}$). Also, when the collection of numbers is given as indices for vectors or matrices, it indicates selecting all indices within the collection. (e.g., $x_{t=[0,T],d=[0,D]} \coloneqq \{\{x_{t,d}\}_{t \in [0:T]}\}_{d \in [0:D]}$). Let $\mathbf{x}_{t,d} \in \mathbb{R}$ the $t$-th observation of the $d$-th feature, and $\mathbf{x}_{[0:T],d}$ and $\mathbf{x}_{[T:T+\tau],d}$ the $d$-th feature's historical inputs and ground truth of future outputs with $T$ and $\tau$ indicating the length of past and future time steps, respectively. Assuming $D$ denotes the number of features, then a stochastic partial-multivariate forecasting model $f$ is formulated as follows:

$$\hat{\mathbf{x}}_{[T:T+\tau],\mathbf{F}} = f(\mathbf{x}_{[0:T],\mathbf{F}}, \mathbf{F}), \qquad \mathbf{F} \sim \mathcal{P}(\mathbf{F}) \quad where \quad \Omega = \{\mathbf{F}|\mathbf{F} \subset [0:D], |\mathbf{F}| = S\}, \quad (1)$$

where $\Omega$ represents the sample space of distribution $\mathcal{P}$. After sampling a subset $\mathbf{F}$ of size $S$ from $\mathcal{P}$, a model $f$ uses the feature indices in $\mathbf{F}$ and their historical observations $\mathbf{x}_{[0:T],\mathbf{F}}$ to forecast the future values of the selected features $\hat{\mathbf{x}}_{[T:T+\tau],\mathbf{F}}$. It is worth noting that this formulation generalizes other forecasting models. Specifically, when $S = 1$, it represents a univariate model; when $1 < S < D$, it corresponds to a partial-multivariate model; and when $S = D$, it describes a complete-multivariate model. Additionally, in partial-multivariate scenarios, when $\mathcal{P}$ is constrained to assign probabilities only to specific subsets within $\Omega$, like Dirac delta distributions, the model becomes deterministic.

## 3.2 STOCHASTIC PARTIAL-MULTIVARIATE TRANSFORMER (SPMFORMER)

For deterministic partial-multivariate or complete-multivariate cases, the architectures are required to capture perpetually unchanging (*i.e.*, static) relationships among features. In other words, $\mathbf{F}$ in equation 1 is always the same throughout training or inference. However, for stochastic partial-multivariate cases, $\mathbf{F}$ can vary when re-sampled, requiring to ability to deal with dynamic relationships. Therefore, inspired by recent Transformer-based models using segmentation (Nie et al., 2023; Zhang & Yan, 2023), we devise SPMformer which addresses this problem by encoding each feature into individual tokens and calculating attention maps only with the feature tokens in $\mathbf{F}$. The overall architecture is illustrated in Figure 2.

After sampling $\mathbf{F}$ in equation 1, the historical observations of selected features $\mathbf{x}_{[0:T],\mathbf{F}} \in \mathbb{R}^{T \times S}$ are encoded into latent tokens $\mathbf{h}^{(0)} \in \mathbb{R}^{N_S \times S \times d_h}$ via a segmentation process where $N_S$ is the number of segments and $d_h$ is hidden size. The segmentation process is formulated as follows:

$$\mathbf{h}_{b,i}^{(0)} = \texttt{Linear}(\mathbf{x}_{[\frac{bT}{N_S}:\frac{(b+1)T}{N_S}],\mathbf{F}_i}) + \mathbf{e}_b^{Time} + \mathbf{e}_{\mathbf{F}_i}^{Feat}, \quad b \in [0, N_S], \quad i \in [0, S], \tag{2}$$

where $\mathbf{F}_i$ denotes the $i$-th element in $\mathbf{F}$. A single linear layer maps observations into latent space with learnable time-wise and feature-wise positional embeddings, $\mathbf{e}^{Time} \in \mathbb{R}^{N_S \times d_h}$ and $\mathbf{e}^{Feat} \in \mathbb{R}^{D \times d_h}$. In most scenarios, we can reasonably assume the input time span $T$ to be divisible by $N_S$ by adjusting $T$ during data pre-processing or padding with zeros as in Zhang & Yan (2023) and Nie et al. (2023).

Subsequently, $\mathbf{h}^{(0)}$ is processed through $L$ SPMformer blocks. Each block is formulated as follows:

$$\bar{\mathbf{h}}^{(\ell-1)} = \mathbf{h}^{(\ell-1)} + \texttt{Feature-Attention}(\mathbf{h}^{(\ell-1)}, \texttt{Temporal-Attention}(\mathbf{h}^{(\ell-1)})), \tag{3}$$

$$\mathbf{h}^{(\ell)} = \bar{\mathbf{h}}^{(\ell-1)} + \texttt{MLP}(\bar{\mathbf{h}}^{(\ell-1)}), \quad \ell = 1, \ldots, L. \tag{4}$$

$\texttt{MLP}$ in equation 4 operates both feature-wise and time-wise, resembling the feed-forward networks found in the original Transformer (Vaswani et al., 2017). As shown in equation 3, there are two types of attention modules:

$$\forall i \in [0:S], \quad \texttt{Temporal-Attention}(\mathbf{h})_{[0:N_S],i} = \texttt{MHSA}(\mathbf{h}_{[0:N_S],i}, \mathbf{h}_{[0:N_S],i}, \mathbf{h}_{[0:N_S],i}), \tag{5}$$

$$\forall b \in [0:N_S], \quad \texttt{Feature-Attention}(\mathbf{h}, \mathbf{v})_{b,[0:S]} = \texttt{MHSA}(\mathbf{h}_{b,[0:S]}, \mathbf{h}_{b,[0:S]}, \mathbf{v}_{b,[0:S]}). \tag{6}$$

$\texttt{MHSA}(\mathbf{Q}, \mathbf{K}, \mathbf{V})$ denotes the multi-head self-attention layer like in Vaswani et al. (2017) where $\mathbf{Q}, \mathbf{K}$, and $\mathbf{V}$ are queries, keys and values. While temporal attention is responsible for capturing temporal dependencies, feature attention mixes representations among features in $\mathbf{F}$.

Starting with initial representations $\mathbf{h}^{(0)}$, SPMformer encoder with $L$ blocks generates final representations $\mathbf{h}^{(L)}$. These representations are then passed through a decoder to forecast future observations. Similar to Nie et al. (2023), the concatenated representations $\mathbf{h}_{[0:N_S],i}^{(L)}$ are mapped to future observations $\mathbf{x}_{[T,T+\tau],\mathbf{F}_i}$ via a single linear layer. For probabilistic forecasting, we replace this decoder with a decoder in Salinas et al. (2019) which takes $\mathbf{h}_{[0:N_S],i}^{(L)}$ as input and outputs the mean and variance of output distributions.

## 3.3 TRAINING ALGORITHM FOR SPMFORMER

To train SPMformer, the process to sample $\mathbf{F}$ from $\mathcal{P}$ is necessary. Ideally, $\mathcal{P}$ should assign higher probabilities to subsets of features that are highly correlated. However, prior knowledge about the relationships between features is usually unavailable. Therefore, we propose a basic form of training algorithm for SPMformer where $\mathcal{P}$ is non-informative (*i.e.*, uniform distribution).[1] In each iteration of training, $N_U$ subsets are sampled from the uniform distribution and SPMformer processes them separately. However, this training algorithm may result in redundancy or omission of some features in each iteration, as some features might be selected multiple times while others might never be chosen across the $N_U$ trials.

---

[1]Despite the lack of prior knowledge, it is advantageous to tailor $\mathcal{P}$ to the dataset using training or other algorithms. However, we propose non-informative cases for two main reasons: *(i)* Since stochastic methods are relatively unexplored, it is essential to first investigate the simplest form of training algorithm using non-informative distributions, and *(ii)* our SPMformer achieves the best performance even with this non-informative distributions. We leave it to future work to find or train optimal distribution $\mathcal{P}$.

To address this issue, we propose a training algorithm based on random partitioning (see Algorithm 1) — note that for-loop in while-loop can be dealt with in parallel with attention masking techniques. In this algorithm, $D$ features are partitioned into $N_U = D/S$ disjoint subsets $\{\mathbf{F}^g\}_{g \in [0:N_U]}$ where $\mathbf{F}^g \subset [0:D], |\mathbf{F}^g| = S, \bigcap_{g \in [0:N_U]} \mathbf{F}^g = \phi, \bigcup_{g \in [0:N_U]} \mathbf{F}^g = [0:D]$ — we assume that $D$ is divisible by $S$. If not, we can handle such cases by repeating some features, as explained in Appendix B. This scheme can minimize the redundancy and omission of features in each iteration. We adopt the training algorithm based on random partitioning as our main training algorithm. Appendix E provides a comparison of these two algorithms in empirical experiments.

---

**Algorithm 1:** Training Algorithm

**Input:** # of features $D$, # of subsets $N_U$, Past obs. $\mathbf{x}_{[0:D]}$, Future obs. $\mathbf{y}_{[0:D]}$

1 **while** *is_converge* **do**
2      Sample all $\mathbf{F}^g$ with random partition;
3      **for** $g \leftarrow 0$ **to** $N_U - 1$ **do**
4          $\mathbf{F} = \mathbf{F}^g$;
5          $\hat{\mathbf{y}}_{\mathbf{F}} = \texttt{SPMformer}(\mathbf{x}_{\mathbf{F}}, \mathbf{F})$;
6          $\text{Loss}_g = \texttt{Loss}(\hat{\mathbf{y}}_{\mathbf{F}}, \mathbf{y}_{\mathbf{F}})$;
         // For-loop is processed in parallel with masked attn.
7      $\text{Loss} = \sum_{g \in [0:N_U]} \text{Loss}_g / N_U$;
8      Train `SPMformer` with Loss;
9 **return** Trained `SPMformer`

---

### 3.4 INFERENCE TECHNIQUE FOR SPMFORMER

After training SPMformer, we can measure inference score using Algorithm 1 without line 8. During inference time, leveraging stochasticity of SPMformer, we sample $\{\mathbf{F}^g\}_{g \in [0:N_U]}$ randomly $N_I$ times, and repeat the inference process $N_I$ times with these sampled subsets, averaging $N_I$ outputs to obtain the final outcomes. In Section 4.3, we observe that this inference technique enhances forecasting performance as $N_I$ increases. It is worth noting that without any additional computation cost (*i.e* $N_I = 1$), SPMformer still achieves state-of-the-art performance against baselines in Appendix F.

Under the assumption that sampling subsets of highly correlated features improves performance, we offer our conjecture on why our inference technique enhances forecasting accuracy. Let $\mathcal{P}(\mathbf{F}_*) = p$ be the probability that we sample a specific subset $\mathbf{F}_*$. Then, the probability of sampling $\mathbf{F}_*$ at least once out of $N_I$ trials is $1 - (1-p)^{N_I}$. Given that $0 \leq p \leq 1$, $1 - (1-p)^{N_I}$ increases as $N_I$ increases. By treating a specific subset $\mathbf{F}_*$ as one that includes mutually significant features, our inference technique with a large $N_I$ increases the likelihood of selecting a subset including highly correlated features at least once, thereby improving forecasting performance.

### 3.5 THEORETICAL ANALYSIS ON SPMFORMER

In this section, we provide theoretical reasons for superiority of our stochastic partial-multivariate models over univariate, complete-multivariate, and deterministic partial-multivariate ones, based on PAC-Bayes framework, similar to other works (Woo et al., 2023; Amit & Meir, 2019; Valle-Pérez & Louis, 2020). Let a neural network $f$ be a stochastic partial-multivariate model which samples subsets $\mathbf{F}$ of $S$ size as defined in equation 1. Also, $\mathcal{T}$ is a training dataset which consists of $m$ instances sampled from the true data distribution. $\mathcal{H}$ denotes the hypothesis class of $f$ with $\mathbf{P}(h)$ and $\mathbf{Q}(h)$ representing the prior and posterior distributions over the hypotheses $h$, respectively. Then, based on McAllester (1999), the generalization bound of $f$ is given by:

**Theorem 1.** *Under some assumptions, with probability at least $1 - \delta$ over the selection of the sample $\mathcal{T}$, we have the following for generalized loss $l(\mathbf{Q})$ under posterior distributions $\mathbf{Q}$.*

$$l(\mathbf{Q}) \leq \sqrt{\frac{-H(\mathbf{Q}) + \log \frac{1}{\delta} + \frac{5}{2} \log m + 8 + C}{2m - 1}}, \tag{7}$$

*where $H(\mathbf{Q})$ is the entropy of $\mathbf{Q}$, (i.e., $H(\mathbf{Q}) = E_{h \sim \mathbf{Q}}[-\log \mathbf{Q}(h)]$) and $C$ is a constant.*

In equation 7, the upper bound depends on $m$ and $-H(\mathbf{Q})$, both of which are related to $S$. Selecting subsets of $S$ size from $D$ features leads to $\binom{D}{S}$ possible cases, affecting $m$ (*i.e.*, $m \propto \binom{D}{S}$). This is because each subset is regarded as a separate instance as in Figure 1. Also, the following theorem reveals relationships between $S$ and $-H(\mathbf{Q})$:

**Theorem 2.** *Let $H(\mathbf{Q}_S)$ be the entropy of a posterior distribution $\mathbf{Q}_S$ with subset size $S$. For $S_+$ and $S_-$ satisfying $S_+ > S_-$. $H(\mathbf{Q}_{S_+}) \leq H(\mathbf{Q}_{S_-})$.*

Theorem 2 is intuitively connected to the fact that capturing dependencies within large subsets of size $S_+$ is usually harder tasks than the case of small $S_-$, because more relationships are captured in the case of $S_+$. As such, the region of hypotheses that satisfies conditions for such hard tasks would be smaller than the one that meets the conditions for a simple task. In other words, probabilities of a posterior distribution $\mathbf{Q}_{S_+}$ might be centered on a smaller region of hypotheses than $\mathbf{Q}_{S_-}$, leading to decreasing the entropy of $\mathbf{Q}_{S_+}$. Refer to Appendix A for full proofs.

Given the unveiled impacts of $S$ on $m$ and $-H(\mathbf{Q})$, we can estimate $S_*$ which is $S$ leading to the lowest upper-bound. When considering only the influence of $m$, $S_*$ is $D/2$, resulting in the largest $\binom{D}{S}$. On the other hand, considering only that of $-H(\mathbf{Q})$, $S_*$ is 1, because $-H(\mathbf{Q})$ decrease as $S$ decreases. Therefore, considering both effects simultaneously, we can think $1 < S_* < D/2$, which means stochastic partial-multivariate models ($1 < S < D$) are better than univariate models ($S = 1$) and complete-multivariate ($S = D$) and the best $S_*$ is between 1 and $D/2$. Furthermore, when comparing stochastic and deterministic partial-multivariate models, stochastic models exhibit a lower generalization bound. This is because stochastic models sample from all $\binom{D}{S}$ possible subsets, while deterministic models are limited to a few predefined subsets, resulting in a lower $m$ compared to the stochastic approach. This analysis is supported by our empirical experimental results in Section 4.3. As of now, since we do not evaluate $H(\mathbf{Q})$ exactly, we cannot compare the magnitudes of effects by $m$ and $-H(\mathbf{Q})$, leaving it for future work. Nevertheless, our analysis from the sign of correlations between $S$ and two factors in the upper-bound still is of importance in that it aligns with our empirical observations.

## 4 EXPERIMENTS

### 4.1 EXPERIMENTAL SETUP

**Datasets.** For long-term and probabilistic forecasting, we use the seven real-world datasets: (*i-iv*) ETTh1, ETTh2, ETTm1, and ETTm2 ($D = 7$), (*v*) Weather ($D = 21$), (*vi*) Electricity ($D = 321$), and (*vii*) Traffic ($D = 862$), similar to previous works (Zhou et al., 2021; Salinas et al., 2019). For each dataset, four settings are constructed with different forecasting lengths $\tau$, which is in {96, 192, 336, 720} with historical length $T = 512$. Also, for short-term forecasting, we use M5 (Makridakis et al., 2022), selecting 100 items randomly in the same store(*i.e.,* $D = 100$) with $T = 256$ and $\tau = 28$ (4 weeks).

**Baselines**. For both long-term and short-term forecasting, we include a variety of models in our baselines. For complete-multivariate baselines, we use Crossformer (Zhang & Yan, 2023), TimesNet (Wu et al., 2023), TSMixer (Chen et al., 2023), DeepTime (Woo et al., 2023), iTransformer (Liu et al., 2024), RLinear (Li et al., 2023), and ModernTCN (donghao & wang xue, 2024). On the univariate side, the baselines include PatchTST (Nie et al., 2023), FITS (Xu et al., 2024), and TimeMixer (Wang et al., 2024). For deterministic partial-multivariate models, we use CAMELOT (Aguiar et al., 2022) as a baseline. In the case of probabilistic forecasting, we include DeepAR (Salinas et al., 2019), ForecasterQR (Wen et al., 2018), and TSDiff (Kollovieh et al., 2023). To further strengthen our baseline set, we also include the top 8 models in long-term forecasting by attaching DeepAR decoders to their last hidden layer.

**Other settings.** For long-term and short-term forecasting, SPMformer is trained with mean squared error (MSE) between ground truth and outputs, whereas we use negative log-likelihood for probabilistic forecasting like Salinas et al. (2019). As evaluation metrics, we report MSE for long-term forecasting, MSE and root mean squared scaled error (RMSSE) for short-term forecasting, and 0.5-risk for probabilistic forecasting, following Zhou et al. (2021); Makridakis et al. (2022); Salinas et al. (2019). For the subset size $S$, we use $S = 3$ for ETT datasets, $S = 7$ for Weather, $S = 30$ for Electricity, $S = 20$ for Traffic, $S = 25$ for M5, satisfying $1 < S < D/2$. Also, for the inference technique of SPMformer, we set $N_I$ to 3. A detailed description of experimental settings is in Appendix C.

### 4.2 FORECASTING RESULT

Table 1, Table 2, and Table 3 show evaluation metrics of representative baselines along with the SPMformer for each task. SPMformer outperforms all baselines in 13 out of 15 cases and achieves

Table 1: MSE in long-term forecasting tasks. For each dataset, scores are averaged over $\tau \in \{96, 192, 336, 720\}$. The best score in each experimental setting is in boldface and the second best is underlined.

| Data | Partial-Multivariate | | Univariate | | | Complete-Multivariate | | | | | | |
|---|---|---|---|---|---|---|---|---|---|---|---|---|
| | SPMformer | CAMELOT | PatchTST | FITS | TimeMixer | Crossformer | TimesNet | TSMixer | DeepTime | iTransformer | RLinear | ModernTCN |
| ETTh1 | **0.392** | 0.405 | 0.413 | 0.406 | 0.411 | 0.570 | 0.487 | 0.412 | 0.423 | 0.479 | 0.409 | 0.404 |
| ETTh2 | 0.322 | 0.324 | 0.331 | 0.333 | **0.316** | 1.618 | 0.383 | 0.355 | 0.434 | 0.383 | 0.328 | 0.322 |
| ETTm1 | **0.343** | 0.356 | 0.353 | 0.358 | 0.348 | 0.427 | 0.422 | 0.347 | 0.354 | 0.407 | 0.359 | 0.351 |
| ETTm2 | **0.248** | 0.253 | 0.256 | 0.254 | 0.256 | 1.001 | 0.331 | 0.267 | 0.259 | 0.291 | 0.253 | 0.253 |
| Weather | **0.217** | 0.233 | 0.226 | 0.221 | 0.222 | 0.231 | 0.258 | 0.225 | 0.238 | 0.244 | 0.243 | 0.224 |
| Electricity | **0.149** | 0.164 | 0.159 | 0.165 | 0.156 | 0.173 | 0.209 | 0.160 | 0.166 | 0.162 | 0.164 | 0.156 |
| Traffic | **0.382** | 0.413 | 0.391 | 0.418 | 0.388 | 0.527 | 0.621 | 0.407 | 0.425 | **0.382** | 0.418 | 0.396 |
| Avg.Rank | **1.143** | 5.571 | 5.571 | 6.286 | 3.571 | 11.000 | 11.143 | 6.000 | 9.000 | 8.286 | 7.000 | 3.143 |

Table 2: RMSSE and MSE in short-term forecasting tasks in M5 when $\tau = 28$ (4 weeks).

| Score | Partial-Multivariate | | Univariate | | | Complete-Multivariate | | | | | | |
|---|---|---|---|---|---|---|---|---|---|---|---|---|
| | SPMformer | CAMELOT | PatchTST | FITS | TimeMixer | Crossformer | TimesNet | TSMixer | DeepTime | iTransformer | RLinear | ModernTCN |
| MSE | **7.418** | 7.481 | 8.695 | 8.327 | 7.529 | 7.676 | 8.157 | 7.995 | 8.186 | 7.421 | 8.176 | 7.518 |
| RMSSE | **0.803** | 0.811 | 0.879 | 0.869 | 0.814 | 0.837 | 0.851 | 0.847 | 0.854 | 0.810 | 0.848 | 0.809 |
| Avg.Rank | **1.000** | 3.500 | 12.000 | 11.000 | 5.000 | 6.000 | 8.500 | 7.000 | 10.000 | 2.500 | 8.500 | 3.000 |

Table 3: 0.5-risk in probabilistic forecasting tasks. For each dataset, scores are averaged over $\tau \in \{96, 192, 336, 720\}$.

| Data | Partial-Multivariate | | Univariate | | | | Complete-Multivariate | | | | | |
|---|---|---|---|---|---|---|---|---|---|---|---|---|
| | SPMformer | CAMELOT | PatchTST | FITS | TimeMixer | TSDiff | TSMixer | iTransformer | RLinear | ModernTCN | DeepAR | ForecasterQR |
| ETTh1 | **0.657** | 1.184 | 1.225 | 0.971 | 0.985 | 1.053 | 0.911 | 0.828 | 0.858 | 0.847 | 1.220 | 1.002 |
| ETTh2 | **0.348** | 0.683 | 0.706 | 0.547 | 0.633 | 0.861 | 0.678 | 0.477 | 0.506 | 0.401 | 1.253 | 0.943 |
| ETTm1 | **0.552** | 1.182 | 1.191 | 0.898 | 0.709 | 0.924 | 0.760 | 0.675 | 0.757 | 0.645 | 1.001 | 0.882 |
| ETTm2 | **0.273** | 0.664 | 0.666 | 0.497 | 0.460 | 0.716 | 0.459 | 0.345 | 0.400 | 0.317 | 0.791 | 0.713 |
| Weather | 0.723 | 1.562 | 1.563 | 1.247 | 1.626 | 0.987 | 0.828 | 1.161 | 1.152 | 0.861 | 0.919 | **0.606** |
| Electricity | **0.391** | 1.035 | 1.034 | 0.831 | 0.495 | 1.313 | 0.491 | 0.510 | 0.516 | 0.518 | 0.621 | 0.482 |
| Traffic | **0.442** | 1.128 | 1.119 | 0.979 | 0.602 | 1.168 | 0.731 | 0.598 | 0.599 | 0.582 | 0.692 | 0.520 |
| Avg.Rank | **1.143** | 9.857 | 10.429 | 7.571 | 6.429 | 9.857 | 5.286 | 4.000 | 5.000 | 3.286 | 9.286 | 5.857 |

the second place in the remaining two. We also provide visualizations of long-term forecasting results of SPMformer and some baselines in Appendix G.2, which shows the superiority of SPMformer. The scores are measured with $N_I = 3$, and in Appendix F, we provide another long-term forecasting result which shows that our SPMformer still outperforms other baselines even with $N_I = 1$. We refer the readers to Appendix G.1 for full results in each $\tau$.

### 4.3 ANALYSIS

In this section, we provide some analysis on our SPMformer. We refer the readers to Appendix G for additional experimental results.

**Empirical result supporting the theoretical analysis.** In Section 3.5, we think that $S_*$ leading to the best forecasting performance is between 1 and $D/2$. To validate this analysis, we provide Table 4, which shows that partial-multivariate settings ($1 < S < D$) outperform others with $S = 1$ or $D$, in most cases. On top of that, our analysis is further supported by the U-shaped plots in Figure 3 where the best MSE is achieved when $1 < S < D/2$ and the worst one is in $S \in \{1, D\}$.

On top of that, to demonstrate that stochastic partial-multivariate models outperform deterministic ones by not being restricted to a few predefined subsets, we conduct an additional experiment, shown in Figure 4, where we vary the size of the subset pool $\mathbf{F}^{all}$ while keeping $S$ fixed. In the original training of SPMformer, the subset pool includes all possible cases, resulting in $\binom{D}{S}$ possible subsets. However, in this experiment, we reduce the pool size to $|\mathbf{F}^{all}| = \alpha \times N_U$ by randomly removing some subsets, where $N_U$ is the number of subsets sampled in each iteration and $\alpha \in \{1, 400, 1600, 6400, \text{Max}\}$. The 'Max' condition corresponds to $\alpha$ yielding the full set of $\binom{D}{S}$ subsets. As shown in Figure 4, the forecasting performance improves as the size of $\alpha$ increases. These experimental results align with our theoretical analyses that stochastic partial-multivariate models achieve better performance by not being constrained to a limited number of predefined cases.

Table 4: Comparison among three types of models by adjusting $S$ in SPMformer. For each dataset of long-term and probabilistic forecasting, scores are averaged over $\tau \in \{96, 192, 336, 720\}$.

| SPMformer Variants | Long-Term Forecasting (MSE) | | | | | Short-Term Forecasting (RMSSE) | Probabilistic Forecasting (0.5-Risk) | | | | |
|---|---|---|---|---|---|---|---|---|---|---|---|
| | ETTh1 | ETTm1 | Weather | Elec. | Traffic | M5 | ETTh1 | ETTm1 | Weather | Elec. | Traffic |
| $S = 1$ | 0.395 | 0.350 | 0.218 | 0.160 | 0.400 | 0.805 | 0.838 | 0.692 | 0.943 | 0.495 | 0.572 |
| $1 < S < D$ | **0.392** | **0.343** | **0.217** | **0.149** | **0.382** | **0.803** | **0.657** | **0.552** | **0.723** | **0.391** | **0.442** |
| $S = D$ | 0.393 | 0.361 | 0.222 | 0.161 | 0.395 | 0.821 | 0.805 | 0.661 | 0.947 | 0.512 | 0.577 |

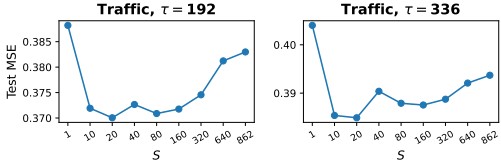

Figure 3: Test MSE by changing $S$.

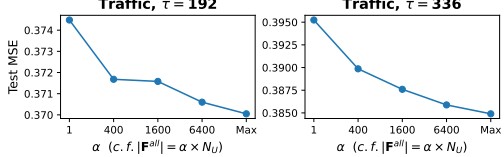

Figure 4: Test MSE by changing $|\mathbf{F}^{all}|$, fixing $S$.

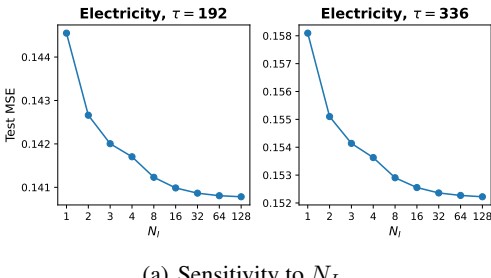

(a) Sensitivity to $N_I$

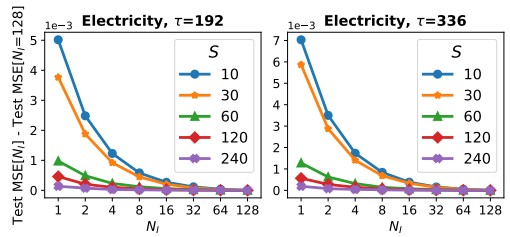

(b) Changes in the effect of $N_I$ when $S$ increases

Figure 5: The effect of $N_I$ on test MSE when (a) $S$ is fixed to the selected hyperparameter and (b) $S$ changes. For (b), the y axis shows the difference of test MSE between when $N_I \in \{1, 2, 4, 8, 16, 32, 64, 128\}$ and $N_I = 128$.

Table 5: MSE of SPMformer with various inference techniques in long-term forecasting — note that all variants of SPMformer are trained with the same algorithms as ours. To identify relevance (significance) of features to others, we utilize attention scores after training SPMformer.

| Inference Technique | Electricity ($D = 321$) | | | | Traffic ($D = 862$) | | | |
|---|---|---|---|---|---|---|---|---|
| | $\tau$=96 | 192 | 336 | 720 | 96 | 192 | 336 | 720 |
| Proposed Technique with $N_I = 3$ (Ours) | **0.125** | **0.142** | **0.154** | **0.176** | **0.345** | **0.370** | **0.385** | **0.426** |
| Sampling A Subset of Mutually Significant Features | 0.132 | 0.148 | 0.174 | 0.205 | 0.352 | 0.372 | 0.386 | 0.428 |
| Sampling A Subset of Mutually Insignificant Features | 0.135 | 0.167 | 0.178 | 0.235 | 0.377 | 0.410 | 0.410 | 0.444 |

**Analysis on the inference technique.** In Section 3.4, we introduce an inference technique that leverages the inherent stochasticity of SPMformer, where the inference process is repeated $N_I$ times, averaging $N_I$ outputs. Figure 5(a) shows the forecasting performance as $N_I$ varies. We observe that Test MSE monotonically decreases as $N_I$ gets large. In Figure 5(b), we investigate relationships between the feature subset size $S$ and $N_I$ by measuring performance gain by increasing $N_I$ in various $S$. This figure shows that the effect of increasing $N_I$ tends to be smaller, as $S$ increases. We think this is because a single subset $\mathbf{F}$ with large $S$ can contain a number of features, so mutually significant features can be included in such large subsets at least once only with few repetitions.

Besides the inference technique based on random selection, we explore another technique which samples subsets of mutually important features by selecting some keys with the highest attention scores per query. We compare this technique to the counterpart which selects keys based on the lowest attention score. In Table 5, we provide the forecasting MSE of each inference technique. — note that only the inference method is different while the training algorithm remains the same as the original one in Algorithm 1. In that an inference technique utilizing the highest attention scores outperforms one with the lowest ones, attention scores are helpful in identifying relationships between features to some extent. Therefore, we think this information will be helpful for approximating true $\mathcal{P}$.

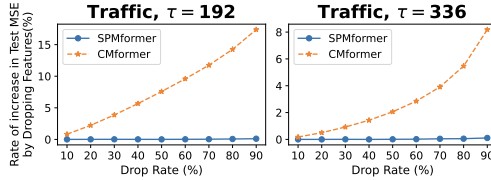 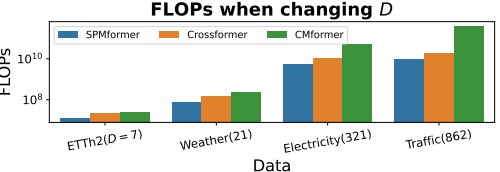

Figure 6: Increasing rate of test MSE by dropping $n\%$ features in SPMformer or Complete-Multivariate Transformer (CMformer).

Figure 7: FLOPs of self-attention for inter-feature dependencies in various Transformers when changing $D$.

**Other advantages of SPMformer.** In the real world, some features in time series are often missing. Inspired by the works that address irregular time series where observations at some time steps (Che et al., 2016; Kidger et al., 2020) are missing, we randomly drop some features of input time series in the inference stage and measure the increasing rate of test MSE in undropped features. For comparison, we use the original SPMformer and a complete-multivariate version of SPMformer (CMformer) by setting $S$ to $D$. SPMformer can address the missingness by simply excluding missing features in the random sampling process, while CMformer has no choice but to pad dropped features with zeros. In Figure 6, unlike the other case, SPMformer maintains its forecasting performance, regardless of the drop rate of the features. This robust characteristic gives SPMformer more applicability in real-world situations where some features are not available.

For Transformers with inter-feature attention modules, we compare the costs of their inter-feature modules using floating point operations (FLOPs) in Figure 7. When naïvely computing inter-feature attention (CMformer), the attention cost is $\mathcal{O}(D^2)$ where $D$ is the number of features. In contrast, due to capturing only partial relationships, the attention cost of SPMformer is reduced to $\mathcal{O}(SD)$ where $S$ is the size of each subset. In Appendix D, we elaborate on the details of the reason why the inter-feature module in SPMformer achieves $\mathcal{O}(SD)$. Given that small $S$ is enough to generate good forecasting performance (*e.g.*, $S = 20\sim30$ for $100\sim800$ features), the attention cost is empirically efficient. As a result, SPMformer achieves the lowest FLOPs compared to others, as shown in Figure 7. Although Crossformer achieves $O(RD)$ complexities with low-rank approximations where $R$ is the rank, our SPMformer shows quite efficient costs, compared to them.

## 5 CONCLUSION

We introduce a new class of multivariate forecasting methods, called *stochastic partial-multivariate methods*, which generalize existing approaches such as univariate, deterministic partial-multivariate, and complete-multivariate methods. As part of this, we develop the SPMformer model. SPMformer first samples clusters (subsets) of a complete feature set from given distributions and captures dependencies only within clusters using a single inter-feature attention module shared by all clusters. Under usual situations without prior knowledge on clustering, we propose a basic form of training algorithm for SPMformer with non-informative clustering distributions. Extensive experiments show that SPMformer outperforms baseline models in long-term, short-term, and probabilistic forecasting tasks. To explain SPMformer's superior performance, we theoretically analyze the upper-bound on generalization errors of SPMformer compared to univariate, deterministic partial-multivariate, and complete-multivariate ones, and provide empirical results supporting the results of the theoretical analysis. Additionally, we enhance forecasting accuracy by introducing a simple inference technique for SPMformer. Finally, we highlight SPMformer's useful characteristics in terms of the efficiency of inter-feature attention and robustness under missing features against complete-multivariate models.

**Future research.** Further theoretical analysis is needed to better explain partial-multivariate models, including more precise calculations of the entropy of posterior distributions and the relaxation of certain assumptions. Additionally, since we have only tested the case where $\mathcal{P}$ is a uniform distribution, future work will focus on identifying the optimal $\mathcal{P}$ for SPMformer. We believe our work could have a positive impact on those developing foundation models for time series due to the following two reasons: *(i)* time series datasets often vary in the number of features, and our feature sampling scheme, where the subset size is always $S$, can address this heterogeneity, and *(ii)* even in cases with an extremely large number of features, our method enables efficient training. Therefore, we plan to test our approach on these heterogeneous and extreme cases.

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

# A PROOF

## A.1 PROOF FOR THEOREM 1

Starting from McAllester's bound on generalization errors (McAllester, 1999), we derive generalization bound in Theorem 1. Before getting into the main part, we define some notations. Let a neural network $f$ be a stochastic partial-multivariate model which samples subsets $\mathbf{F}$ consisting of $S$ features from a complete set of $D$ features as defined in equation 1. $\mathcal{H}$ denotes hypothesis class of $f$, and $\mathbf{P}(h)$ and $\mathbf{Q}(h)$ are a prior and posterior distribution over the hypotheses $h$, respectively. Also, $(\mathbf{x}, \mathbf{y})$ is a input-output pair in an entire dataset and $(\mathbf{x}^{\mathcal{T}}, \mathbf{y}^{\mathcal{T}})$ is a pair in a training dataset $\mathcal{T}$ with $m$ instances sampled from the entire dataset. At last, $\hat{\mathbf{y}} = f(\mathbf{x})$ is the output value of a neural network $f$, and $l(\mathbf{Q})$ and $\hat{l}(\mathbf{Q}, \mathcal{T})$ are generalized and empirical training loss under posterior distributions $\mathbf{Q}$ and training datasets $\mathcal{T}$.

Subsequently, we list assumptions for proof:

**Assumption 1.** *The maximum and minimum values of $\mathbf{y}$ are known and min-max normalization is applied to $\mathbf{y}$ (i.e., $0 \leq \mathbf{y} \leq 1$).*

**Assumption 2.** *The output values of a neural network are assumed to be between 0 and 1, (i.e., $0 \leq \hat{\mathbf{y}} \leq 1$).*

**Assumption 3.** *For posterior distributions $\mathbf{Q}$, $\mathbf{Q}$ is pruned. In other words, we set $\mathbf{Q}(h) = 0$ for hypotheses $h$ where $\mathbf{Q}(h) < \mathbf{P}(h)$ and renormalize it.*

**Assumption 4.** *For any hypothesis $h$, $\mathbf{P}(h) > \omega$ where $\omega$ is the minimum probabilities in $\mathbf{P}(h)$ and $\omega > 0$.*

**Assumption 5.** *For posterior distributions $\mathbf{Q}$ and training datasets $\mathcal{T}$, $\hat{l}(\mathbf{Q}, \mathcal{T}) \approx 0$.*

Given that min-max normalization has been often used in time-series domains with empirical minimum and maximum values (Bhanja & Das, 2019), Assumption 1 can be regarded as a reasonable one. Also, by equipping the last layer with some activation functions such as Sigmoid or Tanh (hyperbolic tangent) like Xu et al. (2019) and adequate post-processing, Assumption 2 can be satisfied.[2] As for Assumption 3, according to (McAllester, 1999), it might have very little effects on $\mathbf{Q}$. Finally, because Transformers can universally approximate any continuous sequence-to-sequence function (Yun et al., 2020), (possibly, extended to general deep neural networks with the universal approximation theorem (Cybenko, 1989)), any hypothesis $h$ can be approximated with proper parameters in $f$. Thus, we can assume $\mathbf{P}(h) > w > 0$ for any $h$ when sampling the initial parameters of $f$ from the whole real-number space (Assmuption 4). Also with proper training process and this universal approximation theorem, $\hat{l}(\mathbf{Q}, \mathcal{T})$ might approximate to zero (Assumption 5). With these assumptions, the proof for Theorem 1 is as follows:

*Proof.* Let MSE be a loss function $l$. Then, according to Assumption 1 and 2, $0 \leq l(h, (\mathbf{x}, \mathbf{y})) \leq 1$ for any data instance $(\mathbf{x}, \mathbf{y})$ and hypothesis $h$. Then, with probability at least $1 - \delta$ over the selection of the sample $\mathcal{T}$ of size $m$, we have the following for $\mathbf{Q}$ (McAllester, 1999):

$$l(\mathbf{Q}) \leq \hat{l}(\mathbf{Q}, \mathcal{T}) + \sqrt{\frac{D(\mathbf{Q}\|\mathbf{P}) + \log\frac{1}{\delta} + \frac{5}{2}\log m + 8}{2m - 1}}, \tag{8}$$

where $D(\mathbf{Q}\|\mathbf{P})$ denotes Kullback-Leibler divergence from distribution $\mathbf{Q}$ to $\mathbf{P}$. Due to Assumption 5, $\hat{l}(\mathbf{Q}, \mathcal{T}) \approx 0$. Also, because $E[\log\frac{1}{\mathbf{P}(h)}] < \log E[\frac{1}{\mathbf{P}(h)}] < \log\frac{1}{\omega} = C$ with Jensen's inequality and Assumption 4, $D(\mathbf{Q}\|\mathbf{P}) = E_{h\sim\mathbf{Q}}[\log\frac{\mathbf{Q}(h)}{\mathbf{P}(h)}] = E[\log\mathbf{Q}(h)] + E[\log\frac{1}{\mathbf{P}(h)}] < E[\log\mathbf{Q}(h)] + C$. Therefore, we can derive Theorem 1 by substituting $\hat{l}(\mathbf{Q}, \mathcal{T})$ and $D(\mathbf{Q}\|\mathbf{P})$ with 0 and $E[\log\mathbf{Q}(h)] + C$, respectively:

$$l(\mathbf{Q}) \leq \sqrt{\frac{E_{h\sim\mathbf{Q}}[\log\mathbf{Q}(h)] + \log\frac{1}{\delta} + \frac{5}{2}\log m + 8 + C}{2m - 1}}. \tag{9}$$

---

[2]Assumption 1 and 2 can be considered somewhat strong but should be satisfied to utilize McAllester's bound widely used for estimating generalization errors (Valle-Pérez & Louis, 2020; Amit & Meir, 2019). When the conditions of McAllester's bound are relaxed, we can also relax our assumptions.

$\square$

Based on this theorem, we provide a theoretical analysis which is the impact of $S$ on $m$ and $-H(Q)$. However, an additional assumption is required to make the rationale valid as follows:

**Assumption 6.** *For the region of hypothesis $h'$ where $\mathbf{Q}(h') > 0$, the prior distribution satisfies $\log \frac{1}{\mathbf{P}(h')} \leq C_{max}$ where $C_{max}$ is small enough to be ignored in upper-bound.*

It is possible that the upper-bound is dominated by $C \to \infty$ when $w \to 0$. As such, $P(h)$ needs to be distributed properly over the region of hypothesis $h'$ where $\mathbf{Q}(h') > 0$ not to result in $C \to \infty$, leading to Assumption 6. This assumption can be satisfied when the prior distribution is non-informative which is natural in Bayesian statistics under the assumption that prior knowledge is unknown (i.e. $P(h) \propto 1$). For any countable set of all possible inputs $\{\mathbf{x}_i\}_{i=1}^N$, probabilities of each $h$ can be represented as $p(h) = \prod_{i=1}^N p(\hat{\mathbf{y}}_i^h|\mathbf{x}_i)$ where $\hat{\mathbf{y}}_i^h = f_h(\mathbf{x}_i)$ is the output of a function $f_h$ under hypothesis $h$ (Domingos, 2012). Because $0 \leq \hat{\mathbf{y}}_i^h \leq 1$ (Assumption 2) and $p(\hat{\mathbf{y}}_i^h|\mathbf{x}_i)$ is a uniform distribution under the non-informative assumption, $p(\hat{\mathbf{y}}_i^h|\mathbf{x}_i) = 1$. As such, the prior distribution under the non-informative assumption is $\mathbf{P}(h) = 1$, leading to $C_{max} = 0$ which is small enough not to dominate upper-bound. On top of that, we can indirectly solve this problem by injecting appropriate inductive biases in the form of architectures or regularizers, which can help to allocate more probability to each hypothesis (*i.e.*, increase $\omega$) by reducing the size of the whole hypothesis space $\mathcal{H}$.

## A.2 PROOF FOR THEOREM 2

To provide a proof for Theorem 2, we first prove Lemma 1. For Lemma 1, we need the following assumption:

**Assumption 7.** *A neural network $f$ models models $p(\mathbf{y}|\mathbf{x})$ where $(\mathbf{x}, \mathbf{y})$ is an input-output pair.*

By regarding the output of a neural network $\hat{\mathbf{y}}$ as mean of normal or Student's $t$-distribution like in Rasul et al. (2024), Assumption 7 can be satisfied. Then, Lemma 1 and a proof are as follows:

**Lemma 1.** *Let $\hat{l}(\mathbf{Q}_S, \mathcal{T}_S)$ be a training loss with posterior distributions $\mathbf{Q}_S$ and a training dataset $\mathcal{T}_S$ when a subset size is $S$. Accordingly, $\hat{l}(\mathbf{Q}_S, \mathcal{T}_S) < \epsilon$ with small $\epsilon$ is a training objective. Then, for $S_+$ and $S_-$ where $S_+ > S_-$, $\mathbf{Q}_{S_+}$ satisfies both $\hat{l}(\mathbf{Q}_{S_+}, \mathcal{T}_{S_+}) < \epsilon$ and $\hat{l}(\mathbf{Q}_{S_+}, \mathcal{T}_{S_-}) < \epsilon$. (On the other hands, $\mathbf{Q}_{S_-}$ is required to satisfy only $\hat{l}(\mathbf{Q}_{S_-}, \mathcal{T}_{S_-}) < \epsilon$.)*

*Proof.* Let $S_+$ and $S_-$ be subset size where $S_+ > S_-$. $\mathbf{F}_{S_-}$ be any subset of $S_-$ size sampled from a complete set of features, and $\mathbf{F}_{S_+}$ is any subset of $S_+$ size among ones that satisfy $\mathbf{F}_{S_-} \subset \mathbf{F}_{S_+}$. $\mathbf{F}_R$ is the set of elements that are in $\mathbf{F}_{S_+}$ but not in $\mathbf{F}_{S_-}$ (*i.e.*, $\mathbf{F}_R = \mathbf{F}_{S_+} - \mathbf{F}_{S_-}$). $\hat{l}(\mathbf{Q}_S, \mathcal{T}_S)$ is a training loss value with posterior distributions $\mathbf{Q}_S$ and a training dataset $\mathcal{T}_S$ when a subset size is $S$. Then, after training process satisfying $\hat{l}(\mathbf{Q}_{S_+}, \mathcal{T}_{S_+}) < \epsilon$ where $\epsilon$ is a small value, we can say that $f$ under $\mathbf{Q}_{S_+}$ outputs the true value of $p(\mathbf{y}_{\mathbf{F}_{S_+}}|\mathbf{x}_{\mathbf{F}_{S_+}})$, according to Assumption 7. In the following process, we demonstrate that $p(\mathbf{y}_{\mathbf{F}_{S_-}}|\mathbf{x}_{\mathbf{F}_{S_-}})$ can be derived from $p(\mathbf{y}_{\mathbf{F}_{S_+}}|\mathbf{x}_{\mathbf{F}_{S_+}}) = p(\mathbf{y}_{\mathbf{F}_{S_-}}, \mathbf{y}_{\mathbf{F}_R}|\mathbf{x}_{\mathbf{F}_{S_-}}, \mathbf{x}_{\mathbf{F}_R})$:

$$\int_{\mathbf{y}_{\mathbf{F}_R}} E_{\mathbf{x}_{\mathbf{F}_R}|\mathbf{x}_{\mathbf{F}_{S_-}}} [p(\mathbf{y}_{\mathbf{F}_{S_-}}, \mathbf{y}_{\mathbf{F}_R}|\mathbf{x}_{\mathbf{F}_{S_-}}, \mathbf{x}_{\mathbf{F}_R})] d\mathbf{y}_{\mathbf{F}_R}, \tag{10}$$

$$= \int_{\mathbf{y}_{\mathbf{F}_R}} \int_{\mathbf{x}_{\mathbf{F}_R}} p(\mathbf{y}_{\mathbf{F}_{S_-}}, \mathbf{y}_{\mathbf{F}_R}|\mathbf{x}_{\mathbf{F}_{S_-}}, \mathbf{x}_{\mathbf{F}_R}) p(\mathbf{x}_{\mathbf{F}_R}|\mathbf{x}_{\mathbf{F}_{S_-}}) d\mathbf{x}_{\mathbf{F}_R} d\mathbf{y}_{\mathbf{F}_R}, \tag{11}$$

$$= p(\mathbf{y}_{\mathbf{F}_{S_-}}|\mathbf{x}_{\mathbf{F}_{S_-}}), \tag{12}$$

In that expectation can be approximated by an empirical mean with sufficient data and integral can be addressed with discretization, we can think that $p(\mathbf{y}_{\mathbf{F}_{S_-}}|\mathbf{x}_{\mathbf{F}_{S_-}})$ can be derived from $p(\mathbf{y}_{\mathbf{F}_{S_+}}|\mathbf{x}_{\mathbf{F}_{S_+}})$. According to this fact, $f$ under $\mathbf{Q}_+$ should be able to output not only true $p(\mathbf{y}_{\mathbf{F}_{S_+}}|\mathbf{x}_{\mathbf{F}_{S_+}})$ but also true $p(\mathbf{y}_{\mathbf{F}_{S_-}}|\mathbf{x}_{\mathbf{F}_{S_-}})$. Therefore, we conclude that $\mathbf{Q}_+$ have to satisfy both $\hat{l}(\mathbf{Q}_{S_+}, \mathcal{T}_{S_+}) < \epsilon$ and $\hat{l}(\mathbf{Q}_{S_+}, \mathcal{T}_{S_-}) < \epsilon$. $\square$

With Lemma 1, we provide a proof for Theorem 2:

*Proof.* Let $h$ be a hypothesis on a space defined when a subset size is $S$. Then, we can denote a posterior distribution which is trained to decrease $\hat{l}(\mathbf{Q}_S, \mathcal{T}_S)$ as follows:

$$\mathbf{Q}(h_S) = p(h_S | c_S = 1), \quad \text{where} \quad c_S = \begin{cases} 1, & \hat{l}(h, \mathcal{T}_S) < \epsilon, \\ 0, & \text{otherwise,} \end{cases} \tag{13}$$

According to Lemma 1, for $S_+$ and $S_-$ where $S_+ > S_-$, the posterior distributions of two cases can be represent as $\mathbf{Q}(h_{S_+}) = p(h_{S_+} | c_{S_+} = 1, c_{S_-} = 1)$ and $\mathbf{Q}(h_{S_-}) = p(h_{S_-} | c_{S_-} = 1)$, respectively. With the following two assumptions, we can prove Theorem 2:

**Assumption 8.** *hypotheses $h_{S_+}$ and $h_{S_-}$ have similar distributions after training with $\mathcal{T}_{S_-}$ (i.e., $p(h_{S_+} | c_{S_-} = 1) \approx p(h_{S_-} | c_{S_-} = 1)$).*

**Assumption 9.** *Prior distributions are nearly non-informative (i.e., $P(h) \propto 1$).*

Assumption 8 can be considered reasonable because we can make the training process of a model of subset size $S_+$ very similar to that of subset size $S_-$ with a minimal change in architecture such as input and output masking. Also, as for Assumption 9, non-informative prior is usually used under usual situations without prior knowledge in Bayesian statistics.

$\mathbf{Q}(h_S)$ can be expanded as $p(h_S | c_S) \propto p(c_S | h_S) p(h_S) \propto p(c_S | h_S)$, according to Assumption 9. Because we exactly know whether to satisfy $\hat{l}(h, \mathcal{T}_S) < \epsilon$ given $h$, $p(c_S | h_S)$ is 1 when a given $h_S$ satisfies $c_S$ or 0, otherwise. Thus, $\mathbf{Q}(h_S)$ are defined as follows:

$$\mathbf{Q}(h_S) = p(h_S | c_S = 1) = \begin{cases} \eta_S, & c_S = 1 \text{ given } h_S, \\ 0, & \text{otherwise,} \end{cases} \tag{14}$$

Similarly, $\mathbf{Q}(h_{S_+})$ and $\mathbf{Q}(h_{S_-})$ can be expanded as $p(h_{S_+} | c_{S_+}, c_{S_-}) \propto p(c_{S_+}, c_{S_-} | h_{S_+}) p(h_{S_+}) \propto p(c_{S_+}, c_{S_-} | h_{S_+})$ and $p(h_{S_-} | c_{S_-}) = p(h_{S_+} | c_{S_-}) \propto p(c_{S_-} | h_{S_+}) p(h_{S_+}) \propto p(c_{S_-} | h_{S_+})$, according to Assumption 8 and 9. A region of hypothesis satisfying both $c_{S_+} = 1$ and $c_{S_-} = 1$ is smaller than that satisfying either of them. Because the probability of $h$ in a region satisfying conditions has the same value and $\int_h p(h) dh = 1$ is maintained, $h$ in the small region is allocated higher probabilities than $h$ in the large one. Therefore, $\eta_{S_+} > \eta_{S_-}$ and the entropy $H(\mathbf{Q}_{S_-})$ is larger than $H(\mathbf{Q}_{S_+})$:

$$\square$$

So far, we have finished a proof for Theorem 2. We additionally provide Theorem 3 which is a variant of Theorem 2 where Assumption 9 can be relaxed while proposing the relationships between $H(\mathbf{Q}_{S_-})$ and $H(\mathbf{Q}_{S_+})$ in the expectation level:

**Theorem 3.** *for $S_+$ and $S_-$ satisfying $S_+ > S_-$, $H(\mathbf{Q}_{S_+}) \leq H(\mathbf{Q}_{S_-})$ in expectation over $c_{S_+}$.*

*Proof.* Let $\tilde{h}_{S_+}$ be the $h_{S_+} | c_{S_-} = 1$ (*i.e.*, $\mathbf{Q}(h_{S_+}) = p(h_{S_+} | c_{S_+} = 1, c_{S_-} = 1) = p(\tilde{h}_{S_+} | c_{S_+} = 1)$). Then, $H(\tilde{h}_{S_+} | c_{S_+})$ can be expanded as follows:

$$H(p(\tilde{h}_{S_+} | c_{S_+})), \tag{15}$$

$$= H(p(c_{S_+} | \tilde{h}_{S_+})) + H(p(\tilde{h}_{S_+})) - H(p(c_{S_+})), \tag{16}$$

$(\because \text{Bayes' rule for conditional entropy states}),$

$$= H(p(\tilde{h}_{S_+})) - H(p(c_{S_+})) \tag{17}$$

$(\because \text{when } h \text{ is given, we know whether to satisfy } \hat{l}(h, \mathcal{T}_{S_+}) < \epsilon. \text{ (i.e., } H(p(c_{S_+} | \tilde{h}_{S_+})) = 0),$

From this expansion, we can derive $H(p(\tilde{h}_{S_+} | c_{S_+})) \leq H(p(\tilde{h}_{S_+}))$ because entropy of $p(c_{S_+})$ must be larger than 0 (*i.e.*, $H(p(c_{S_+})) \geq 0$). By substituting $H(\mathbf{Q}_{S_-})$ for $H(p(\tilde{h}_{S_+}))$ according to Assumption 8 and $E_{c_{S_+}}[H(\mathbf{Q}_{S_+})]$ for $H(p(\tilde{h}_{S_+} | c_{S_+}))$, we can derive Theorem 3.

Also, based on Chebyshev's inequality, we can calculate the least probabilities at which $H(\mathbf{Q}_{S_+}) < H(\mathbf{Q}_{S_-})$ are satisfied, given the variance $\sigma^2 = Var_{c_{S_+}}[H(p(\tilde{h}_{S_+}|c_{S_+}))]$:

$$p\left[H(\mathbf{Q}_{S_+}) < H(\mathbf{Q}_{S_-})\right] \quad \leq \quad 1 - \frac{\sigma^2}{(H(p(c_{S_+})))^2} \tag{18}$$

$\square$

## B  How to Handle Non-Divisible Cases of SPMformer with Random Partitioning

In this section, we further elaborate on how to deal with the cases where the number of features $D$ is not divisible by the size of subsets $S$. We simply repeat some randomly chosen features and augment them to the original input time series, in order to make the total number of features divisible by $S$. After finishing the forecasting procedure with the augmented inputs, we drop augmented features from outputs. The details are delineated in Algorithm 2.

---

**Algorithm 2:** How to handle non-divisible cases of SPMformer with random partitioning

---

**Input:** # of features $D$, Subset size $S$, Past obs. $\mathbf{x}_{[0:D]}$

1  $\mathbf{V} = \{0, 1, ..., D-1\}; \quad N_U = \lceil\frac{D}{S}\rceil; \quad R = D \% S;$

2  **if** $R \neq 0$ **then**

3  $\quad$ Randomly split $\mathbf{V}$ into $\mathbf{V}^+, \mathbf{V}^-$, where $|\mathbf{V}^+| = D - R, |\mathbf{V}^-| = R, \mathbf{V}^+ \cap \mathbf{V}^- = \phi;$

4  $\quad$ Get $\{\mathbf{F}^g\}_{g \in [0, N_U - 1]}$ by randomly partitioning $\mathbf{V}^+$;

5  $\quad$ $\mathbf{V}^{++} = \{v_i | v_i \text{ is a random sample from } \mathbf{V}^+ \text{ without replacement}, i = [0, S - R]\};$

6  $\quad$ $\mathbf{F}^{N_U - 1} = \mathbf{V}^- \cup \mathbf{V}^{++}$

7  **else**

8  $\quad$ Get $\{\mathbf{F}^g\}_{g \in [0, N_U]}$ by randomly partitioning $\mathbf{V}$;

9  **for** $g \leftarrow 0$ **to** $N_U - 1$ **do**

10  $\quad$ $\hat{\mathbf{y}}_{\mathbf{F}^g} = \text{SPMformer}(\mathbf{x}_{\mathbf{F}^g}, \mathbf{F}^g);$

11  **if** $R \neq 0$ **then**

12  $\quad$ Remove features of $\mathbf{V}^{++}$ from $\hat{\mathbf{y}}_{\mathbf{F}^{N_U - 1}};$

13  Sort $\{\hat{\mathbf{y}}_{\mathbf{F}^g}\}_{g \in [0, N_U]}$ by feature index and get $\hat{\mathbf{y}}_{[0:D]};$

14  **return** Predicted future observations $\hat{\mathbf{y}}_{[0:D]};$

---

## C  Details of Experimental Environments

We conduct experiments on this software and hardware environments: PYTHON 3.7.12, PYTORCH 2.0.1, and NVIDIA GEFORCE RTX 3090.

### C.1  Datasets

We evaluate SPMformer on 8 benchmark datasets for time series forecasting with multiple variables. The normalization and train/val/test splits are also the same with ModernTCN (donghao & wang xue, 2024) which is our main baseline. The information of each dataset is as follows:

- **(1-2) ETTh1,2**[3] (Electricity Transformer Temperature-hourly): They have 7 indicators in the electric power long-term deployment, such as oil temperature and 6 power load features. This data is collected for 2 years and the granularity is 1 hour. Different numbers denote different counties in China. The number of time steps is 17,420.
- **(3-4) ETTm1,2** (Electricity Transformer Temperature-minutely): This dataset is exactly the same with **ETTh1,2**, except for granularity. The granularity of these cases is 15 minutes. The number of time steps is 69,680.

---

[3]https://github.com/zhouhaoyi/ETDataset

- **(5) Weather**[4]: It has 21 indicators of weather including temperature, humidity, precipitation, and air pressure. It was recorded for 2020, and the granularity is 10 minutes. The number of time steps is 52,696.
- **(6) Electricity**[5]: In this dataset, information about hourly energy consumption from 2012 to 2014 is collected. Each feature means the electricity consumption of one client, and there are 321 clients in total. The number of time steps is 26,304.
- **(7) Traffic**[6]: Traffic dataset pertains to road occupancy rates. It encompasses hourly data collected by 862 sensors deployed on San Francisco freeways during the period spanning from 2015 to 2016. The number of time steps is 17,544.
- **(8) M5**[7]: The M5 dataset is used in the M5 Forecasting Competition, which aims to evaluate and compare different forecasting methods. The competition centers around predicting sales data for a range of products, stores, and timeframes. We randomly select 100 items for our task. The number of time steps is 1,907.

## C.2 HYPERPARAMETERS

The details of hyperparameters used in the SPMformer are delineated in this section. For the number of segments $N_S$, we use $N_S = 32$ for M5, 8 for Traffic, and 64 for others. The dropout ratio $r_{\text{dropout}}$ is in {0.1, 0.2, 0.3, 0.4, 0.7}. The hidden dimension $d_h$ is in {32,64,128,256,512}. The number of heads in self-attention $n_h$ is in {2,4,8,16} and the number of layers $L$ is in {1,2,3}. $d_{ff}$ is the hidden size of feed-forward networks in each SPMformer layer and in {32,64,128,256,512}. Also, batch size is 128, 128, 16, and 12 for ETT, Weather, Electricity, and Traffic datasets, respectively. Finally, we set the learning rate and training epochs to $10^{-3}$ and 100, respectively. Finally, we use Adam optimizer to train our model. The selected best hyperparameters of SPMformer are in Table 6.

## D COMPLEXITY ANALYSIS OF INTER-FEATURE ATTENTION IN SPMFORMER

In this section, we elaborate on the reason why the theoretical complexity of inter-feature attention in SPMformer is $\mathcal{O}(SD)$ where $D$ is the number of features and $S$ is the subset size. Attention cost in each subset is $\mathcal{O}(S^2)$. Because random partitioning generates $N_U \approx \frac{D}{S}$ subsets, the final complexity is $N_U \mathcal{O}(S^2) = \frac{D}{S}\mathcal{O}(S^2) = \mathcal{O}(SD)$.

## E THE EFFECT OF TRAINING SPMFORMER WITH RANDOM SAMPLING OR PARTITIONING

In this section, we provide the experimental results where we train SPMformer using a training algorithm with random sampling or partitioning. As shown in Table 7, these two ways are comparable in terms of forecasting performance — note that we adopt the training algorithm based on random partitioning for our main experiments.

## F THE PERFORMANCE OF SPMFORMER WITH $N_I = 1$

In Table 8, we conduct the main experiments including SPMformer with $N_I = 1$ which is the number of repeating an inference process. In this experiment, we include some baselines showing decent forecasting performance. As Table 8 shows, despite $N_I = 1$, SPMformer still gives better results than baselines.

---

[4] https://www.bgc-jena.mpg.de/wetter/
[5] https://archive.ics.uci.edu/ml/datasets/ElectricityLoadDiagrams20112014
[6] http://pems.dot.ca.gov
[7] https://www.kaggle.com/competitions/m5-forecasting-accuracy

Table 6: Selected hyperparameters of SPMformer.

| Data | $\tau$ | $r_{\text{dropout}}$ | $d_h$ | $n_h$ | $L$ | $d_{ff}$ |
|------|--------|---------------------|-------|-------|-----|----------|
| ETTh1 | 96 | 0.7 | 128 | 4 | 1 | 256 |
| | 192 | 0.7 | 32 | 4 | 1 | 256 |
| | 336 | 0.7 | 64 | 8 | 1 | 64 |
| | 336 | 0.7 | 64 | 8 | 1 | 64 |
| ETTh2 | 96 | 0.7 | 512 | 4 | 1 | 256 |
| | 192 | 0.7 | 512 | 2 | 1 | 256 |
| | 336 | 0.7 | 64 | 16 | 1 | 256 |
| | 720 | 0.7 | 64 | 16 | 1 | 128 |
| ETTm1 | 96 | 0.2 | 256 | 2 | 2 | 256 |
| | 192 | 0.1 | 64 | 8 | 1 | 128 |
| | 336 | 0.2 | 64 | 2 | 2 | 64 |
| | 720 | 0.7 | 64 | 4 | 1 | 128 |
| ETTm2 | 96 | 0.7 | 512 | 2 | 1 | 64 |
| | 192 | 0.7 | 128 | 4 | 1 | 32 |
| | 336 | 0.4 | 128 | 2 | 1 | 32 |
| | 720 | 0.7 | 256 | 4 | 1 | 32 |
| Weather | 96 | 0.2 | 128 | 8 | 3 | 256 |
| | 192 | 0.2 | 128 | 16 | 3 | 256 |
| | 336 | 0.4 | 128 | 16 | 3 | 512 |
| | 720 | 0.4 | 128 | 2 | 1 | 256 |
| Electricity | 96 | 0.3 | 256 | 8 | 1 | 256 |
| | 192 | 0.2 | 256 | 4 | 2 | 256 |
| | 336 | 0.2 | 128 | 4 | 3 | 256 |
| | 720 | 0.2 | 256 | 4 | 3 | 256 |
| Traffic | 96 | 0.2 | 512 | 2 | 3 | 512 |
| | 192 | 0.1 | 256 | 4 | 3 | 512 |
| | 336 | 0.2 | 256 | 2 | 3 | 256 |
| | 720 | 0.2 | 512 | 4 | 3 | 512 |
| M5 | | 0.0 | 128 | 8 | 2 | 128 |

Table 7: MSE of training SPMformer using a training algorithm with random sampling or partitioning

| Training Algorithm | ETTh1 ($D = 7$) | | | | ETTh2 (7) | | | | ETTm1 (7) | | | | ETTm2 (7) | | | |
|--------------------|------|------|------|------|------|------|------|------|------|------|------|------|------|------|------|------|
| | 96 | 192 | 336 | 720 | 96 | 192 | 336 | 720 | 96 | 192 | 336 | 720 | 96 | 192 | 336 | 720 |
| Random Partitioning | **0.361** | **0.396** | **0.400** | **0.412** | **0.269** | **0.323** | **0.317** | **0.370** | **0.282** | **0.325** | **0.352** | **0.401** | **0.160** | **0.213** | **0.262** | **0.336** |
| Random Sampling | 0.362 | 0.397 | **0.400** | **0.412** | 0.273 | **0.323** | **0.317** | 0.371 | 0.283 | **0.325** | **0.352** | 0.403 | 0.162 | 0.214 | 0.263 | 0.337 |

| Training Algorithm | Weather (21) | | | | Electricity (321) | | | | Traffic (862) | | | | Avg. Rank |
|--------------------|------|------|------|------|------|------|------|------|------|------|------|------|-----------|
| | 96 | 192 | 336 | 720 | 96 | 192 | 336 | 720 | 96 | 192 | 336 | 720 | |
| Random Partitioning | **0.142** | 0.185 | **0.235** | **0.305** | **0.125** | 0.142 | **0.154** | 0.176 | **0.345** | **0.370** | 0.385 | **0.426** | **1.071** |
| Random Sampling | **0.142** | **0.184** | 0.237 | **0.305** | 0.126 | **0.141** | **0.154** | 0.180 | 0.347 | **0.370** | 0.386 | 0.427 | 1.571 |

# G ADDITIONAL EXPERIMENTS

## G.1 ADDITIONAL EXPERIMENTAL RESULTS IN TABULAR FORMS

In this section, we provide full results for existing experiments. Table 9 and Table 10 are additional results for Table 1 and Table 3, respectively. Also, both Table 11 and Table 12 are for Table 4.

## G.2 ADDITIONAL VISUALIZATION

Like Appendix G.1, this section provides additional visualizations with other datasets or models for existing ones. Figure 8 is for Figure 3, Figure 9 for Figure 4, Figure 10 for Figure 5(a), Figure 11 for Figure 5(b), and Figure 12 for Figure 6. Furthermore, Figure 13 shows the forecasting results of SPMformer, PatchTST, and Crossformer. We select these baselines because they have similar architecture to SPMformer, such as segmentation or inter-feature attention modules. Our method captures temporal dynamics better than baselines.

Table 8: MSE of main forecasting results including SPMformer wiht $N_I = 1$.

| Method | ETTh1 ($D = 7$) | | | | ETTh2 (7) | | | | ETTm1 (7) | | | | ETTm2 (7) | | | |
|---|---|---|---|---|---|---|---|---|---|---|---|---|---|---|---|---|
| | 96 | 192 | 336 | 720 | 96 | 192 | 336 | 720 | 96 | 192 | 336 | 720 | 96 | 192 | 336 | 720 |
| SPMformer | **0.361** | **0.393** | 0.404 | **0.412** | 0.270 | 0.328 | 0.321 | 0.371 | **0.286** | 0.328 | **0.354** | 0.418 | 0.165 | 0.219 | **0.271** | 0.357 |
| CAMELOT | 0.367 | 0.396 | 0.410 | 0.448 | 0.269 | 0.333 | 0.321 | 0.374 | 0.298 | 0.338 | 0.372 | 0.417 | **0.164** | **0.218** | 0.272 | 0.358 |
| TimeMixer | **0.361** | 0.409 | 0.430 | 0.445 | 0.271 | **0.317** | 0.332 | **0.342** | 0.291 | **0.327** | 0.360 | **0.415** | 0.164 | 0.223 | 0.279 | 0.359 |
| ModernTCN | 0.368 | 0.405 | **0.391** | 0.450 | **0.263** | 0.320 | **0.313** | 0.392 | 0.292 | 0.332 | 0.365 | 0.416 | 0.166 | 0.222 | 0.272 | **0.351** |

| Method | Weather (21) | | | | Electricity (321) | | | | Traffic (862) | | | | Avg. Rank |
|---|---|---|---|---|---|---|---|---|---|---|---|---|---|
| | 96 | 192 | 336 | 720 | 96 | 192 | 336 | 720 | 96 | 192 | 336 | 720 | |
| SPMformer | **0.142** | **0.185** | **0.235** | **0.305** | **0.125** | 0.142 | **0.154** | **0.176** | **0.345** | **0.370** | **0.385** | **0.426** | **1.607** |
| CAMELOT | 0.158 | 0.204 | 0.253 | 0.317 | 0.138 | 0.150 | 0.165 | 0.204 | 0.390 | 0.402 | 0.411 | 0.449 | 3.321 |
| TimeMixer | 0.147 | 0.189 | 0.241 | 0.310 | 0.129 | **0.140** | 0.161 | 0.194 | 0.360 | 0.375 | **0.385** | 0.430 | 2.321 |
| ModernTCN | 0.149 | 0.196 | 0.238 | 0.314 | 0.129 | 0.143 | 0.161 | 0.191 | 0.368 | 0.379 | 0.397 | 0.440 | 2.607 |

Table 9: MSE in long-term forecasting tasks. (Additional results for Table 1)

| Data | | Partial-Multivariate | | Univariate | | | Complete-Multivariate | | | | | | |
|---|---|---|---|---|---|---|---|---|---|---|---|---|---|
| | | SPMformer | CAMELOT | PatchTST | FITS | TimeMixer | Crossformer | TimesNet | TSMixer | DeepTime | iTransformer | RLinear | ModernTCN |
| ETTh1 | 96 | **0.361** | 0.367 | 0.370 | 0.372 | **0.361** | 0.427 | 0.465 | **0.361** | 0.372 | 0.396 | 0.364 | 0.368 |
| | 192 | 0.396 | **0.396** | 0.413 | 0.405 | 0.409 | 0.537 | 0.493 | 0.404 | 0.405 | 0.425 | 0.402 | 0.405 |
| | 336 | 0.400 | 0.410 | 0.422 | 0.420 | 0.430 | 0.651 | 0.456 | 0.420 | 0.437 | 0.459 | 0.419 | 0.391 |
| | 720 | **0.412** | 0.448 | 0.447 | 0.426 | 0.445 | 0.664 | 0.533 | 0.463 | 0.477 | 0.638 | 0.451 | 0.450 |
| ETTh2 | 96 | 0.269 | 0.269 | 0.274 | 0.271 | 0.271 | 0.720 | 0.381 | 0.274 | 0.291 | 0.300 | **0.255** | 0.263 |
| | 192 | 0.328 | 0.333 | 0.341 | 0.330 | 0.317 | 1.121 | 0.416 | 0.339 | 0.403 | 0.382 | **0.316** | 0.320 |
| | 336 | 0.320 | 0.321 | 0.329 | 0.353 | 0.332 | 1.524 | 0.363 | 0.361 | 0.466 | 0.424 | 0.325 | **0.313** |
| | 720 | 0.370 | 0.374 | 0.379 | 0.378 | **0.342** | 3.106 | 0.371 | 0.445 | 0.576 | 0.426 | 0.415 | 0.392 |
| ETTm1 | 96 | **0.282** | 0.298 | 0.293 | 0.307 | 0.291 | 0.336 | 0.343 | 0.285 | 0.311 | 0.341 | 0.310 | 0.292 |
| | 192 | **0.325** | 0.338 | 0.333 | 0.338 | 0.327 | 0.387 | 0.381 | 0.327 | 0.339 | 0.381 | 0.337 | 0.332 |
| | 336 | **0.352** | 0.372 | 0.369 | 0.368 | 0.360 | 0.431 | 0.436 | 0.356 | 0.366 | 0.419 | 0.369 | 0.365 |
| | 720 | 0.412 | 0.417 | 0.416 | 0.421 | 0.415 | 0.555 | 0.527 | 0.419 | **0.400** | 0.486 | 0.419 | 0.416 |
| ETTm2 | 96 | **0.163** | **0.164** | 0.166 | 0.165 | 0.164 | 0.338 | 0.218 | **0.163** | 0.165 | 0.184 | **0.163** | 0.166 |
| | 192 | **0.216** | 0.218 | 0.223 | 0.219 | 0.223 | 0.567 | 0.282 | **0.216** | 0.222 | 0.253 | 0.219 | 0.222 |
| | 336 | **0.266** | 0.272 | 0.274 | 0.272 | 0.279 | 1.050 | 0.378 | 0.268 | 0.278 | 0.315 | 0.272 | 0.272 |
| | 720 | **0.349** | 0.358 | 0.361 | 0.359 | 0.359 | 2.049 | 0.444 | 0.420 | 0.369 | 0.412 | 0.360 | 0.351 |
| Weather | 96 | **0.142** | 0.158 | 0.149 | 0.144 | 0.147 | 0.150 | 0.179 | 0.145 | 0.169 | 0.171 | 0.171 | 0.149 |
| | 192 | **0.185** | 0.204 | 0.194 | 0.188 | 0.189 | 0.200 | 0.230 | 0.191 | 0.211 | 0.212 | 0.216 | 0.196 |
| | 336 | **0.235** | 0.253 | 0.245 | 0.239 | 0.241 | 0.263 | 0.276 | 0.242 | 0.255 | 0.260 | 0.261 | 0.238 |
| | 720 | **0.305** | 0.317 | 0.314 | 0.312 | 0.310 | 0.310 | 0.347 | 0.320 | 0.318 | 0.334 | 0.323 | 0.314 |
| Electricity | 96 | **0.125** | 0.138 | 0.129 | 0.137 | 0.129 | 0.135 | 0.186 | 0.131 | 0.139 | 0.132 | 0.136 | 0.129 |
| | 192 | 0.142 | 0.150 | 0.147 | 0.151 | **0.140** | 0.158 | 0.208 | 0.151 | 0.154 | 0.152 | 0.150 | 0.143 |
| | 336 | **0.154** | 0.165 | 0.163 | 0.167 | 0.161 | 0.177 | 0.210 | 0.161 | 0.169 | 0.170 | 0.166 | 0.161 |
| | 720 | **0.176** | 0.204 | 0.197 | 0.206 | 0.194 | 0.222 | 0.231 | 0.197 | 0.201 | 0.192 | 0.206 | 0.191 |
| Traffic | 96 | **0.345** | 0.390 | 0.360 | 0.396 | 0.360 | 0.481 | 0.599 | 0.376 | 0.401 | 0.353 | 0.395 | 0.368 |
| | 192 | **0.370** | 0.402 | 0.379 | 0.408 | 0.375 | 0.509 | 0.612 | 0.397 | 0.413 | 0.370 | 0.407 | 0.379 |
| | 336 | 0.385 | 0.411 | 0.392 | 0.417 | 0.385 | 0.534 | 0.618 | 0.413 | 0.425 | **0.384** | 0.416 | 0.397 |
| | 720 | 0.426 | 0.449 | 0.432 | 0.453 | 0.430 | 0.585 | 0.654 | 0.444 | 0.462 | **0.419** | 0.453 | 0.440 |
| Avg.Rank | | **1.500** | 5.750 | 5.607 | 6.071 | 3.714 | 10.679 | 11.071 | 5.071 | 8.393 | 8.321 | 6.464 | 4.036 |

Table 10: 0.5-risk in probabilistic forecasting tasks. (Additional results for Table 3)

| Data | | Partial-Multivariate | | Univariate | | | | Complete-Multivariate | | | | | |
|---|---|---|---|---|---|---|---|---|---|---|---|---|---|
| | | SPMformer | CAMELOT | PatchTST | FITS | TimeMixer | TSDiff | TSMixer | iTransformer | RLinear | ModernTCN | DeepAR | ForecasterQR |
| ETTh1 | 96 | **0.587** | 1.170 | 1.200 | 0.944 | 0.768 | 1.001 | 0.755 | 0.722 | 0.775 | 0.781 | 1.174 | 0.930 |
| | 192 | **0.648** | 1.177 | 1.196 | 0.963 | 0.761 | 1.052 | 0.937 | 0.839 | 0.825 | 0.842 | 1.119 | 1.010 |
| | 336 | **0.668** | 1.176 | 1.208 | 0.974 | 1.158 | 1.087 | 0.930 | 0.837 | 0.868 | 0.862 | 1.251 | 0.977 |
| | 720 | **0.724** | 1.212 | 1.294 | 1.005 | 1.254 | 1.071 | 1.021 | 0.912 | 0.965 | 0.902 | 1.338 | 1.091 |
| ETTh2 | 96 | **0.297** | 0.665 | 0.675 | 0.530 | 0.572 | 0.789 | 0.617 | 0.396 | 0.438 | 0.354 | 1.343 | 1.026 |
| | 192 | **0.326** | 0.680 | 0.707 | 0.540 | 0.974 | 0.926 | 0.702 | 0.391 | 0.482 | 0.395 | 1.435 | 0.888 |
| | 336 | **0.349** | 0.683 | 0.708 | 0.547 | 0.419 | 0.874 | 0.639 | 0.482 | 0.542 | 0.429 | 1.114 | 1.007 |
| | 720 | **0.419** | 0.704 | 0.736 | 0.571 | 0.569 | 0.853 | 0.756 | 0.639 | 0.562 | 0.426 | 1.121 | 0.850 |
| ETTm1 | 96 | **0.483** | 1.176 | 1.191 | 0.852 | 0.626 | 0.861 | 0.658 | 0.545 | 0.697 | 0.559 | 0.840 | 0.797 |
| | 192 | **0.546** | 1.189 | 1.189 | 0.889 | 0.711 | 0.939 | 0.742 | 0.676 | 0.736 | 0.597 | 1.007 | 0.844 |
| | 336 | **0.561** | 1.178 | 1.187 | 0.914 | 0.704 | 0.914 | 0.799 | 0.703 | 0.773 | 0.603 | 1.033 | 0.932 |
| | 720 | **0.620** | 1.184 | 1.198 | 0.936 | 0.796 | 0.979 | 0.841 | 0.775 | 0.821 | 0.821 | 1.125 | 0.957 |
| ETTm2 | 96 | **0.215** | 0.644 | 0.647 | 0.444 | 0.302 | 0.553 | 0.347 | 0.288 | 0.315 | 0.309 | 0.573 | 0.417 |
| | 192 | **0.260** | 0.654 | 0.658 | 0.488 | 0.457 | 0.688 | 0.414 | 0.316 | 0.372 | 0.306 | 0.744 | 0.674 |
| | 336 | **0.290** | 0.667 | 0.670 | 0.514 | 0.592 | 0.697 | 0.509 | 0.366 | 0.422 | 0.313 | 0.733 | 0.800 |
| | 720 | **0.327** | 0.689 | 0.688 | 0.540 | 0.489 | 0.927 | 0.564 | 0.411 | 0.493 | 0.338 | 1.113 | 0.962 |
| Weather | 96 | 0.595 | 1.554 | 1.548 | 1.210 | 2.235 | 0.844 | 0.650 | 0.984 | 1.079 | 0.748 | 0.761 | **0.496** |
| | 192 | 0.694 | 1.561 | 1.561 | 1.231 | 1.142 | 0.985 | 0.752 | 1.107 | 1.133 | 0.812 | 0.869 | **0.598** |
| | 336 | 0.751 | 1.560 | 1.565 | 1.256 | 1.243 | 1.020 | 0.864 | 1.201 | 1.171 | 0.886 | 0.971 | **0.654** |
| | 720 | 0.853 | 1.571 | 1.577 | 1.289 | 1.883 | 1.098 | 1.047 | 1.350 | 1.225 | 0.997 | 1.075 | **0.676** |
| Electricity | 96 | **0.348** | 1.025 | 1.022 | 0.803 | 0.436 | 1.353 | 0.448 | 0.443 | 0.469 | 0.505 | 0.602 | 0.462 |
| | 192 | **0.376** | 1.030 | 1.029 | 0.818 | 0.462 | 1.319 | 0.486 | 0.475 | 0.492 | 0.510 | 0.635 | 0.482 |
| | 336 | **0.401** | 1.036 | 1.035 | 0.836 | 0.504 | 1.289 | 0.507 | 0.535 | 0.522 | 0.516 | 0.634 | 0.488 |
| | 720 | **0.440** | 1.051 | 1.050 | 0.867 | 0.576 | 1.289 | 0.524 | 0.587 | 0.579 | 0.543 | 0.613 | 0.495 |
| Traffic | 96 | **0.426** | 1.131 | 1.120 | 0.965 | 0.602 | 1.171 | 0.663 | 0.555 | 0.586 | 0.575 | 0.630 | 0.531 |
| | 192 | **0.439** | 1.126 | 1.116 | 0.967 | 0.556 | 1.173 | 0.728 | 0.590 | 0.590 | 0.581 | 0.703 | 0.453 |
| | 336 | 0.446 | 1.125 | 1.115 | 0.976 | 0.629 | 1.173 | 0.760 | 0.619 | 0.597 | 0.575 | 0.709 | **0.441** |
| | 720 | **0.459** | 1.132 | 1.125 | 1.006 | 0.621 | 1.156 | 0.773 | 0.626 | 0.625 | 0.596 | 0.728 | 0.653 |
| Avg.Rank | | **1.179** | 10.036 | 10.321 | 7.643 | 5.536 | 9.643 | 5.607 | 4.179 | 5.071 | 3.500 | 9.143 | 6.071 |

Table 11: MSE of three types of models by adjusting $S$ of SPMformer in long-term forecasting tasks. (Additional results for Table 4)

| SPMformer Variants | ETTh1 ($D = 7$) | | | | ETTh2 (7) | | | | ETTm1 (7) | | | | ETTm2 (7) | | | |
|---|---|---|---|---|---|---|---|---|---|---|---|---|---|---|---|---|
| | 96 | 192 | 336 | 720 | 96 | 192 | 336 | 720 | 96 | 192 | 336 | 720 | 96 | 192 | 336 | 720 |
| $S = 1$ | **0.361** | **0.393** | 0.404 | 0.420 | 0.272 | 0.325 | 0.318 | 0.371 | 0.288 | 0.335 | 0.358 | 0.403 | 0.161 | **0.213** | 0.265 | 0.338 |
| $1 < S < D$ | **0.361** | 0.396 | **0.400** | **0.412** | **0.269** | 0.323 | 0.317 | 0.370 | 0.282 | 0.325 | 0.352 | 0.401 | 0.160 | 0.213 | 0.262 | 0.336 |
| $S = D$ | **0.361** | 0.395 | 0.401 | 0.413 | **0.269** | 0.325 | 0.318 | 0.371 | 0.299 | 0.350 | 0.377 | 0.402 | 0.161 | 0.213 | 0.265 | 0.338 |

| SPMformer Variants | Weather (21) | | | | Electricity (321) | | | | Traffic (862) | | | | Avg. Rank |
|---|---|---|---|---|---|---|---|---|---|---|---|---|---|
| | 96 | 192 | 336 | 720 | 96 | 192 | 336 | 720 | 96 | 192 | 336 | 720 | |
| $S = 1$ | **0.141** | 0.186 | 0.237 | 0.308 | 0.128 | 0.146 | 0.163 | 0.204 | 0.368 | 0.388 | 0.404 | 0.441 | 2.286 |
| $1 < S < D$ | 0.142 | **0.185** | **0.235** | **0.305** | **0.125** | **0.142** | 0.154 | 0.176 | **0.345** | 0.370 | 0.385 | 0.426 | **1.107** |
| $S = D$ | 0.146 | 0.192 | 0.244 | 0.307 | 0.129 | 0.147 | 0.163 | 0.204 | 0.363 | 0.383 | 0.394 | 0.441 | 2.250 |

Table 12: 0.5-risk of three types of models by adjusting $S$ of SPMformer in probabilistic forecasting tasks. (Additional results for Table 4)

| SPMformer Variants | ETTh1 ($D = 7$) | | | | ETTh2 (7) | | | | ETTm1 (7) | | | | ETTm2 (7) | | | |
|---|---|---|---|---|---|---|---|---|---|---|---|---|---|---|---|---|
| | 96 | 192 | 336 | 720 | 96 | 192 | 336 | 720 | 96 | 192 | 336 | 720 | 96 | 192 | 336 | 720 |
| $S = 1$ | 0.750 | 0.815 | 0.847 | 0.939 | 0.387 | 0.385 | 0.438 | 0.572 | 0.580 | 0.685 | 0.718 | 0.783 | 0.269 | 0.337 | 0.388 | 0.431 |
| $1 < S < D$ | **0.587** | **0.648** | **0.668** | **0.724** | **0.297** | **0.326** | **0.349** | **0.419** | **0.483** | **0.546** | **0.561** | **0.620** | **0.215** | **0.260** | **0.290** | **0.327** |
| $S = D$ | 0.733 | 0.798 | 0.811 | 0.878 | 0.361 | 0.375 | 0.423 | 0.561 | 0.536 | 0.662 | 0.655 | 0.790 | 0.239 | 0.327 | 0.391 | 0.400 |

| SPMformer Variants | Weather (21) | | | | Electricity (321) | | | | Traffic (862) | | | | Avg. Rank |
|---|---|---|---|---|---|---|---|---|---|---|---|---|---|
| | 96 | 192 | 336 | 720 | 96 | 192 | 336 | 720 | 96 | 192 | 336 | 720 | |
| $S = 1$ | 0.772 | 0.878 | 0.964 | 1.157 | 0.449 | 0.467 | 0.504 | 0.558 | 0.549 | 0.565 | 0.578 | 0.596 | 2.571 |
| $1 < S < D$ | **0.595** | **0.694** | **0.751** | **0.853** | **0.348** | **0.376** | **0.401** | **0.440** | **0.426** | **0.439** | **0.446** | **0.459** | **1.000** |
| $S = D$ | 0.775 | 0.902 | 0.966 | 1.144 | 0.456 | 0.488 | 0.526 | 0.580 | 0.553 | 0.565 | 0.578 | 0.611 | 2.429 |

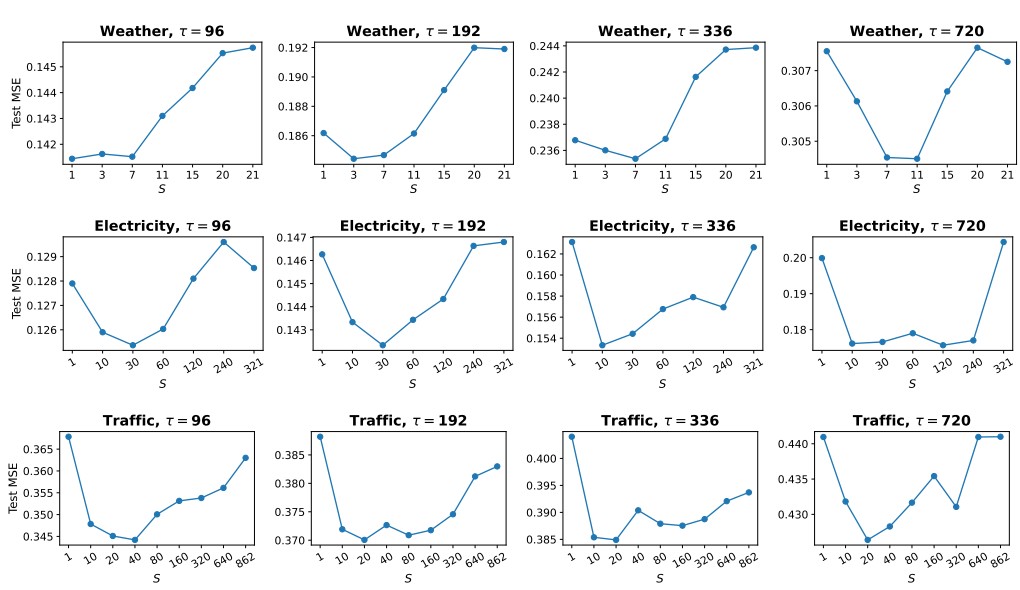

Figure 8: Sensitivity to $S$. (Additional results for Figure 3)

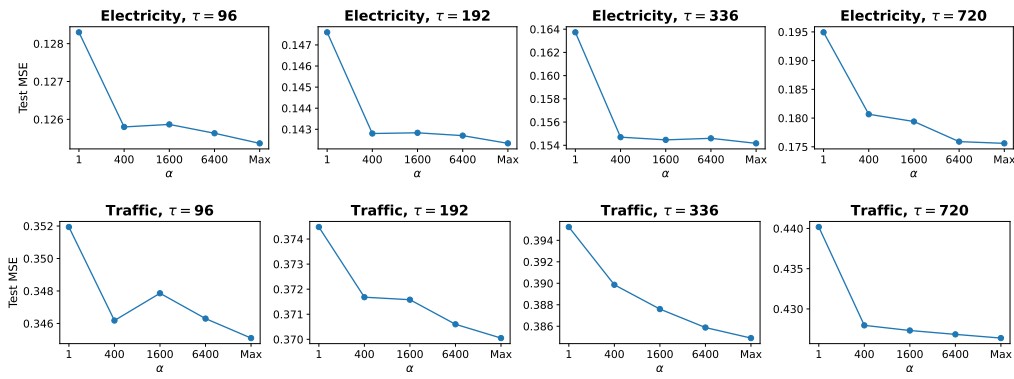

Figure 9: Sensitivity to $|\mathbf{F}^{all}| = \alpha \times N_U$. (Additional results for Figure 4)

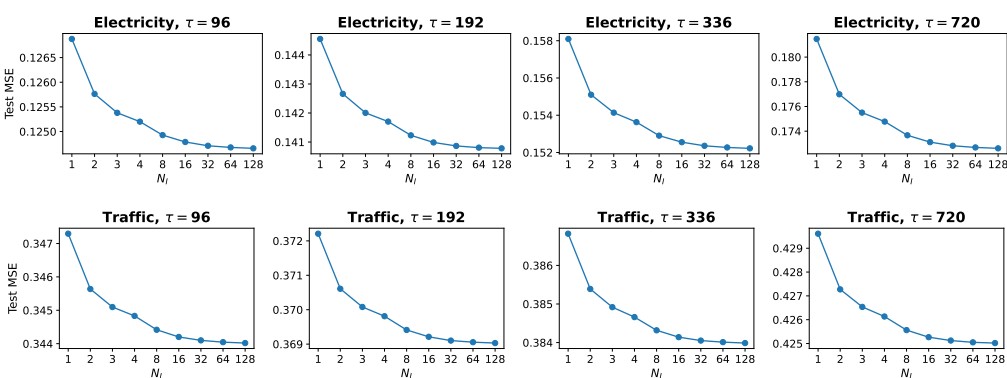

Figure 10: Sensitivity to $N_I$. (Additional results for Figure 5(a))

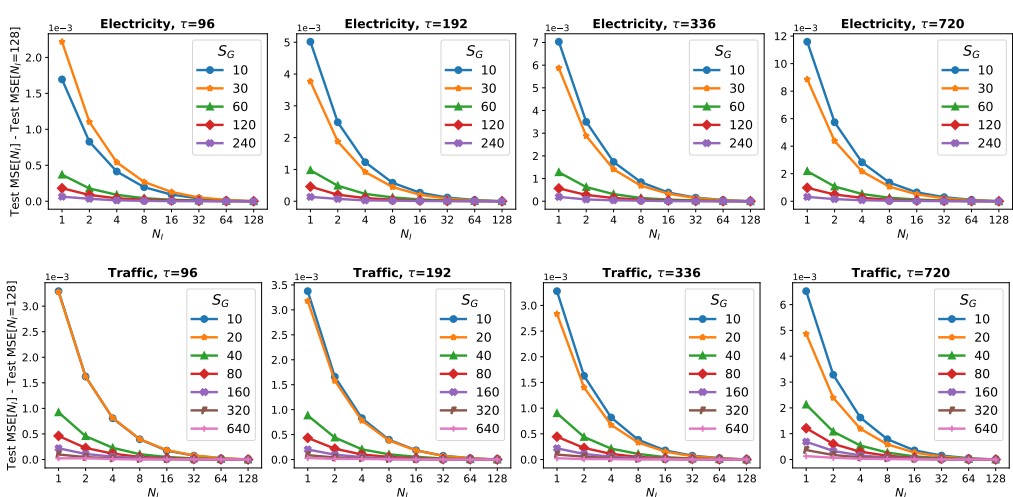

Figure 11: Changes in the effect of $N_I$ on forecasting performance when $S$ increases. (Additional results for Figure 5(b))

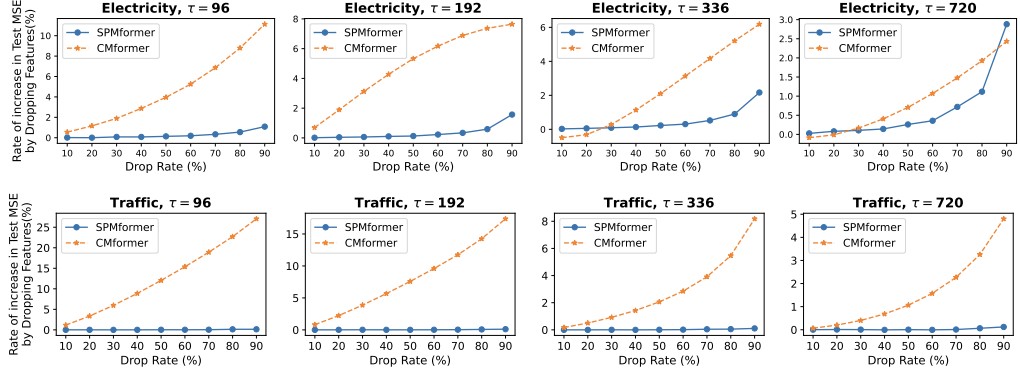

Figure 12: Increasing rate of test MSE by dropping $n\%$ features in SPMformer or Complete-Multivariate Transformer (CMformer). (Additional results for Figure 6)

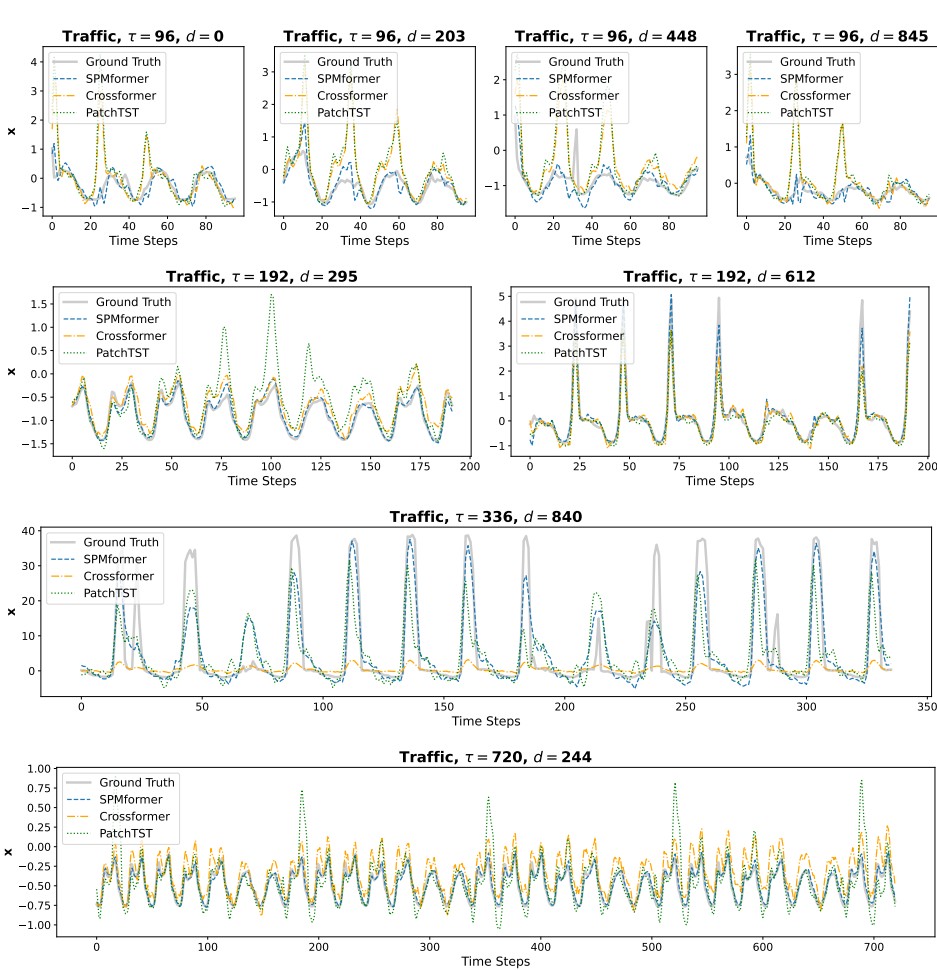

Figure 13: Forecasting results of various segment-based transformers (Crossformer, PatchTST, and SPMformer). Dotted lines and dotted-dashed lines denote baselines, dashed lines denote SPMformer, and solid lines denote ground truth. $\tau$ denotes the length of time steps in future outputs and $d$ denotes a feature index.

