# OpenReview forum: "Stochastically Capturing Partial Relationship among Features for Multivariate Forecasting"
_ICLR.cc/2025/Conference — ICLR 2025 Conference Withdrawn Submission_

### Official Review · Reviewer_81rP · 2024-10-22

**Soundness:** 3
**Presentation:** 3
**Contribution:** 2
**Rating:** 5
**Confidence:** 4

**Summary:**

The paper introduces a novel approach for time-series forecasting called “Stochastic Partial-Multivariate methods,” which generalizes existing models such as univariate, deterministic partial-multivariate, and complete-multivariate methods. The key contribution is the development of the SPMformer model, a Transformer-based architecture that captures relationships among feature subsets stochastically, improving accuracy across various forecasting tasks. The authors provide both theoretical and empirical evidence demonstrating the superiority of SPMformer in long-term, short-term, and probabilistic forecasting. Additionally, the paper highlights SPMformer’s robustness under missing features and its efficiency in handling inter-feature relationships compared to complete-multivariate models.

**Strengths:**

1. This paper introduces a stochastic element into partial-multivariate modeling, it opens up new avenues for more flexible, accurate forecasting methods to handle complex inter-feature dependencies.

2. This paper is well-structured and the methodology is clearly explained. The empirical evaluation is comprehensive.

**Weaknesses:**

1. This paper lacks a detailed rationale for why randomly selecting feature subsets from multivariate time series can enhance forecasting performance. Specifically, if the training and testing sets consist of different feature subsets, it is unclear how this method ensures good generalization performance across unseen data.
2. Although the authors have limited their analysis to non-informative distributions, many real-world time series datasets possess inherent hierarchical structures (e.g., cash flow, population data). Applying random permutation operations in such cases may disrupt these intrinsic relationships, potentially leading to suboptimal performance.
3. The theoretical analysis does not sufficiently explain why random feature subsampling would improve generalization performance.

**Questions:**

See the weaknesses.

---

### Official Review · Reviewer_QSH4 · 2024-10-30

**Soundness:** 1
**Presentation:** 3
**Contribution:** 3
**Rating:** 3
**Confidence:** 5

**Summary:**

The paper proposes SPMFormer, a transformer-based model for multivariate time-series forecasting that is trained on multiple subsets of features, such that the final model runs faster and is more robust to missing values than full multivariate architectures, and that improves upon univariate methods. Feature subsets of size $S$ are sampled uniformly and model's outputs are averaged over several runs $N_I$ at inference.

**Strengths:**

1. **The work is well motivated** in the introduction: provide **a flexible model that can ingest different number and type of time-series** at inference time. I believe that such flexibility **is the main contribution of this work** and that it is key to deploying transformer-based models in practice. Indeed, end users might want to condition forecasting on different sets of time series, depending on their availability or on budget considerations.

2. Extensive experimental results are reported to validate the proposed method for varying $S$, $N_I$ and forecast horizons, and in comparison with several deep learning state-of-the-art techniques. In particular, (i) the results for varying S (e.g., Figure 3) suggest that there exist a sweet spot between running the model on small or large number of features; (ii) Figure 6 clearly illustrates the **robustness of SPMformer to missing values**.

3. The work is generally presented clearly. In particular Section 3.2 is easy to follow and provides the essential architectural details.

**Weaknesses:**

In order of gravity:
1. There are **two flaws in the provided theoretical results**:
- (major) **Theorem 1**, as it is derived from the PAC-Bayesian classical results of [McAllester 1991], **does not hold for time-series data**, because their training sets are not I.I.D. . Indeed the bound stands in probability over independent draws of samples of m instances, while time-series instances notoriously show time dependencies.
- (minor) The conjecture of Section 3.4 about why results improve with the number of inference runs $N_I$ is not supported by the reasoning of lines 241-244 because the derived probability of picking a subset increases not only for highly correlated subsets of features, but for any subset of features, including subsets of unrelated features which, if drawn more often, should result in decreased results, as argued in lines 49-50.

2. **Standard errors are not reported** for any result. Without them, it's impossible to judge the significance of the improvements, hence to support the claim of e.g., lines 23-25,  that "SPMformer outperforms baselines".

3. **The comparison between methods based on FLOPs reported in Figure 7 is not fair**, because it accounts only for the number of operations needed to run a forward pass for a single subset of features (line 453). Because SPMformer requires running the model several times (number of subsets times number of trials), a fair comparison would account for the overall operations required for a full inference pass.

4. The paper does not mention how the hyper-parameters have been selected.

5. The 0.5-risk is not used in the papers [Zhou 2021, Makridakis 2022] as mentioned in line 315.

**Questions:**

1. My understanding is that the subset size S is fixed at training time and then the model is tested with the same S (based on Equation 1, although it is not clearly confirmed in the experimental setup). I think that to unlock the full potential of SPMformer, it would be interesting to show that the model can be trained and run for different values of S without loss of performance, or that the model can generalize to new S, i.e., if trained for a S, it still has good performance on smaller/larger S.

---

### Official Review · Reviewer_NKqB · 2024-10-31

**Soundness:** 3
**Presentation:** 3
**Contribution:** 3
**Rating:** 5
**Confidence:** 3

**Summary:**

This paper studies a partial-multivariate method for capturing inter-feature information for forecasting problems. Existing methods cannot accommodate more complex scenarios where features may be grouped in various ways since groupings are fixed once when they are determined through some specific procedures. The authors, therefore, propose a transformer-based model termed SPMformer to address the challenges. The authors show that the proposed model outperforms various baselines in various forecasting tasks (long-term, short-term, and probabilistic forecasting). Further, they provide a theoretical rationale and empirical analysis for its superiority.

**Strengths:**

The authors provide detailed descriptions of the proposed model. Further, the authors provide a theoretical study based on the proposed model and provide sufficient experiments to compare the performance of the proposed model with other baselines.

**Weaknesses:**

The mathematical notations are not clear enough, which reduces the readability of the paper and the overall score of this paper. Below are some questions on the notations:
1. It is very confusing to write $\bf{F} \sim \mathcal{P}(\bf{F})$ in Eqn. (1). Is $\bf{F}$ a subset or a probability value when inputting to $f$ to get the estimated value?
2. What is the dimension of $\bf{h} _ {b,i}^{(0)}$? How do one concatenate $\bf{h} _ {b,i}^{(0)}$ to form $\bf{h}^{(0)}$?
3. What are the dimensions of $\bf{e} _ {b}^{Time}$ and $\bf{e} _ {\bf{F} _ i}^{Feature}$ in Eqn. (2)? It seems that the vectors $\bf{e} _ {b}^{Time}$ and $\bf{e} _ {\bf{F} _ i}^{Feature}$ are different and addition is impossible. Further, how do one concatenate $\bf{e} _ {b}^{Time}$ ($\bf{e} _ {\bf{F} _ i}^{Feature}$) to form $\bf{e}^{Time}$  ($\bf{e}^{Feature}$)?
4. Is Eqn. (7) valid for any arbitrary loss functions $l$? Please justify in the paper.

**Questions:**

The score will be adjusted if the authors can alleviate my doubts about the questions stated below:
1. To my realization, $S$ should be updated during the training process. Otherwise, the proposed model should still be classified as a deterministic partial multivariate case. Please explain how to update it during training in Algorithm 1.
2. Authors claim that the proposed algorithm addresses the issue of omission of some features. For instance, suppose D=3 with three features F1, F2, F3, and S=2. According to the proposed algorithm, splitting gives two sets: {F1, F2} and {F3}. Following the algorithm given in the Appendix, {F3} should be extended; let’s say it is extended to {F2, F3}. To my understanding, it is possible that {F1, F3} may contribute a lot to prediction and the existing algorithm my ignore it. Please explain.
3. Could you provide a comparison between the proposed method and the existing algorithms on the computation time? The proposed model captures the partial relationship stochastically, the running time may be much larger than existing baselines according to my realizations. There should be a trade-off between the predicted accuracy and the computation time.

---

### Official Review · Reviewer_LvW5 · 2024-11-05

**Soundness:** 2
**Presentation:** 2
**Contribution:** 2
**Rating:** 3
**Confidence:** 4

**Summary:**

This paper introduces a partial-multivariate method that randomly samples variable subsets from multivariate time series during training and inference. The authors compared the proposed method to several baselines.

**Strengths:**

1. The presentation is generally easy to follow.
2. The experiment section provides details.

**Weaknesses:**

1. The motivation for the proposed method has unclear points. In the introduction section, the authors mention that  "methods often face limitations due to their deterministic approach to grouping." The deterministic approach is not necessarily limited in this regard if it manages to capture the underlying stable pattern in data. Even if this is the case, it is incomprehensible how randomly sampling variable subsets can solve the issue, and meanwhile, without principled sampling and an understanding of data patterns, the proposed randomized method introduces additional issues like missing important variables and ignoring inter-relations across variables. The example of stock is opposite to the motivation, because sectors and trading regions/markets of stocks are mostly static, and market caps are slowly changing.

   Essentially, the introduction does not justify the necessity and rationale of randomizing the input variables of prediction models during training and inference, nor does it recognize the additional issues introduced by randomizing input variables.

2. The proposed method (i.e., from Eq.1 to 6) is mostly applied, with little technical contribution and new insights. In Eq.1, the distribution $\mathcal{P}$ is not formulated throughout the paper and is eventually realized by a uniform distribution, making the method less principled.

3. In lines 166 to 167,  it is mentioned that "However, for stochastic partial multivariate cases, F can vary when re-sampled, requiring to ability to deal with dynamic relationships". This claim is ambiguous and depends on the definition of dynamic relationships. If the dynamic relationship is at the instance level, varying input variables for training and inference can hardly handle this case. The authors should be cautious about overclaims.

4. In Eq.2, the segmentation and tokenization process seems to share the linear projection. As the input variables change and their patterns might differ significantly during training and inference, sharing the projection in Eq.2 might be suboptimal.

5. In line 197, does "the concatenated representations" lead to extremely high dimensions if it is the product of the timestep and the hidden dimension?

6. In line 7 of Algorithm 1, the averaged loss is confusing, since each element corresponds to a different set of input variables.

7. Sec. 3.4 discusses the number of samples for inference, however, what also matters for inference is the variable sample size. In the Other Settings of the experiment section, the sample size is dataset-specific, and there is no guidance on how to set it.

8. The theoretical analysis rarely answers the rationale of the method. Theorem 2, as presented in lines 270-289, primarily boils down to the impact of sample size and instance number, which is rather superficial and does not provide meaningful insight into the method’s effectiveness.

**Questions:**

See above.

---

### Author Response · Authors · 2024-11-18
**Common Response to All Reviewers**

We sincerely appreciate the reviewers’ constructive feedback and thoughtful insights. After careful deliberation, we have decided to withdraw our submission. However, we would like to take this opportunity to clarify our current contributions and outline the key improvements suggested by the reviewers to further refine our work.

**Summary of Contributions.** Our work introduces the concept of stochastic partial-multivariate models to the time-series literature for the first time:
- These models capture stochastic partial relationships between features in multivariate time-series.
- They are designed to naturally accommodate scenarios where sub-clusters of features exhibit diverse and dynamic relationships.

**Technical Contributions.** Our primary contributions to stochastic partial-multivariate models are as follows:
- **Formulation of Stochastic Partial-Multivariate Models (Section 3.1):** We formulate stochastic partial-multivariate models.
- **Implementation with Self-Attention (Section 3.2):** To address dynamic feature relationships, we propose SPMformer, which incorporates self-attention modules. SPMformer encodes each feature as an individual token and computes attention maps within subsets of features, effectively managing partial relationships.
- **Training and Inference Techniques (Sections 3.3–3.4):** In cases where prior knowledge about feature relationships is unavailable, we introduce training algorithms based on random sampling and partitioning. Additionally, we develop resource-intensive but performance-enhancing inference techniques tailored for SPMformer.
- **Theoretical Analysis (Section 3.5):** Using the PAC-Bayes framework, we provide theoretical analysis to demonstrate the advantages of SPMformer over traditional univariate and fully multivariate models.

Based on the reviewers' constructive feedback, we plan to improve our work by clearly articulating the motivations and formulations of our approach [LvW5], addressing ambiguous or unclear statements [LvW5, NKqB, 81rP], refining our theoretical analysis [LvW5, QSH4], designing experiments with improved fairness [QSH4], and providing more comprehensive experimental details [QSH4].

Thank you once again for your valuable feedback and support.
Sincerely,
Authors of Submission1703.

---

### Note · Authors · 2024-11-18

I have read and agree with the venue's withdrawal policy on behalf of myself and my co-authors.